

# Retention During Freezing of Raindrops, Part I: Investigation of Single and Binary Mixtures

Martanda Gautam[1], Alexander Theis[2], Jackson Seymore[1], Moritz Hey[1], Stephan Bormann[1,2], Karoline Diehl[1], Subir K. Mitra[2], and Miklós Szakáll[1]

[1]Institute for Atmospheric Physics, Johannes Gutenberg University, Mainz, Germany
[2]Particle Chemistry Department, Max Plank Institute for Chemistry, Mainz, Germany

**Correspondence:** Martanda Gautam (mgautam@uni-mainz.de) and Miklós Szakáll (szakall@uni-mainz.de)

**Abstract.** The interaction with freezing processes and vertical transport of trace gases into the upper atmosphere during deep convection is critical to understanding the distribution of aerosol precursors and their climate effects. We conducted experimental studies inside a walk-in cold room for freely levitating rain drops (D = 2 mm) using an acoustic levitator apparatus. We investigated the effect of freezing raindrops on the retention of organic species for the first time with silver iodide as the ice nucleating agent. Quantitative chemical analysis determined the retention coefficient, which is defined as the fraction of a chemical species remaining in the ice phase compared to their initial liquid phase concentrations. We measured the retention coefficients of nitric acid, formic acid, acetic acid, and 2-nitrophenol as single components. Furthermore, we determined the retention coefficients of these substances as binary mixtures. Our results show the dominance of physical properties over their chemical counterparts on overall retention for the investigated large drops. Thus, for rain sized drops almost everything is fully retained during the freezing process, even for species with low effective Henry's law constants. An ice shell is formed within $4.8$ ms around the drops just after the freezing was initiated. This ice shell formation was found to be the controlling factor for the overall retention of the investigated species, which inhibited any further expulsion of dissolved substances from the drop.

## 1 Introduction

The Earth's atmosphere consists of a diverse range of chemical constituents, starting from ever present gases such as nitrogen, oxygen, carbon-dioxide, ozone, etc., to a wide range of chemicals in trace amounts as well. Biogenic and anthropogenic source contributors are known to be important for understanding the role that trace constituents have on the atmosphere over long timescales (Kolb et al., 2010; Andreae, 2019). However, vertical redistribution can be just as critical (Martini et al., 2011; Ervens, 2015; Wang et al., 2016). During convective transport, there is a rapid redistribution of trace gases and aerosols from boundary layer to the upper troposphere, which can alter the overall concentration of the chemical constituents (Warneck, 1999; Corti et al., 2008; Ervens, 2015).

Organic aerosol mass is usually underestimated in the boundary layer and beyond (Carlton et al., 2009; Hodzic et al., 2020). As a consequence, the potential impact of aerosols on the global radiation budget, radiative forcing, and overall climate can be misrepresented (Lohmann and Feichter, 2005; Tsigaridis et al., 2014; Shrivastava et al., 2017; Sporre et al., 2020). Williamson





et al. (2019) also reported that there is an under-representation of total organic mass due to low estimations for new particle
formation, particularly in tropical convective regions.

During vertical transport in deep convective systems, there is an evident phase change of the water droplets as they undergo
cooling and subsequent freezing at lower temperature regimes higher up in the atmosphere. Trace gases dissolved in these drops
could be either retained, revolatized, or scavenged during the freezing process (Pruppacher and Klett, 2010). The fraction of
chemical species remaining inside the frozen drop, compared to their initial concentration in liquid phase before freezing,
results in the so-called retention coefficient. Substances that are completely retained after freezing will have a retention co-
efficient of 1. Modelling studies (Mari et al., 2000; Barth et al., 2001, 2007; Tost et al., 2010; Long et al., 2010; Bela et al.,
2016; Cuchiara et al., 2020; Ryu and Min, 2022; Cuchiara et al., 2023) concerning convective transport and redistribution of
trace gases have stressed on the importance of experimentally determined retention coefficients. However, such experimental
databases are quite few in this regard.

Previous studies on experimentally determining retention coefficients in context of riming of supercooled droplets of single
substances (Iribarne et al., 1983; Lamb and Blumenstein, 1987; Iribarne et al., 1990; Snider et al., 1992; Snider and Huang,
1998; von Blohn et al., 2011, 2013; Jost et al., 2017; Borchers et al., 2024) help bridge the uncertainty gap and provide
a backbone for effective parameterization for modelling frameworks. The term "riming-retention" will be used to refer to
these above mentioned studies collectively. The following substances were studied for retention during riming of supercooled
droplets: $SO_2$, $H_2O_2$, $O_2$, $HNO_3$, HCl, $NH_4$, formic acid, acetic acid, malonic acid, oxalic acid, formaldehyde, $\alpha$-pinene
oxidation derivatives and nitro- aromatic compounds. These experimental studies revealed dependencies of the retention of
trace gases on both chemical and physical properties. Additionally, there is established correlation with effective Henry's law
coefficient ($H^*$) and a retention indicator (RI) parameter, which relates experimentally derived retention coefficients to model
derived values (Stuart and Jacobson, 2003, 2004). $H^*$ shows the dependence on the solubility and dissociative properties of
trace gases, whereas RI provides a ratio of expulsion timescales to freezing timescales. A freezing time significantly lower
than the solute expulsion time would result in a chemical substance being retained. These expulsion timescales are described in
Schwartz (1986), that take into account the aqueous, interfacial, and gaseous mass transfer rates and the aqueous phase kinetics
as explained in Jost et al. (2017). In addition to these chemical properties, physical properties such as drop size, ventilation
around the drop, temperature, and liquid water content are the major contributing factors affecting retention (Jost et al., 2017;
Jost, 2017). The above mentioned experimental studies concerning riming-retention were mostly related with cloud sized
droplets (i.e. diameters in the μm size range), for which the chemical properties were determined to be the dominant factors. The
present study focuses on large rain drops (diameters in the mm size range), which have not been experimentally investigated
thus far. Freezing of raindrops are especially important for the case of convective clouds with warm bases where collision and
coalescence can produce such large mm sized drops, which can be further transported into the upper troposphere during deep
convection. Henceforth, the term "freezing-retention" will be used to refer to the present study, investigating retention during
freezing of rain drops.

The motivation for this study was to investigate and understand the retention of chemical species dissolved in larger drops,
and thereby augment experimental databases to further enhance modeling frameworks. To conceptualize our experimental

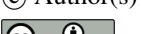



outlook, we selected four chemical substances with increasing $H^*$ values, namely: 2-nitrophenol, acetic acid, formic acid, and

nitric acid. These substances are commonly found in the atmosphere and their previously measured retention coefficient values

for riming with cloud droplet sizes lie between 0 to 1 and scale with $H^*$. In addition to investigating these four substances

as single components, we also studied their potential interactions as binary mixtures. Binary mixtures were studied to infer a

more systematic understanding of the retention process as in how the differential incorporation or segregation of two substances

during freezing might affect their overall retention.

## 2  Methods

### 2.1  Experimental Setup

In this study, we used the Mainz-Acoustic Levitator (M-AL) setup (Fig. 1), which was placed inside a walk-in cold room.

M-AL employs an ultrasonic wave source (58 KHz) and a metal reflector to produce a standing wave. Water drops can be

injected with a syringe and levitated contact-free at the intersection of the incident and reflected waves (i.e. at the nodes of the

standing wave). The diameters of the levitated water drops used in this study were $2.0 \pm 0.1$ mm. The M-AL is enclosed inside

a Plexiglas housing to minimize any external interference to the standing wave. More details about the M-AL can be found in

Diehl et al. (2014) and Szakáll et al. (2021).

In addition to the ultrasonic source, an infrared thermometer (KT 19.82 II, Heitronics) was used to measure the surface

temperature of the levitated drops, and a USB camera (USB-103H, Phytec GmbH, Germany) to record the drop size informa-

tion. The top left section of the schematic (Fig. 1) shows the placement of the video camera, which had a wide video graphics

array of $752 \times 480$ pixels and a minimum pixel size of $6 \times 6$ μm. The infrared thermometer can be seen at the bottom right

section of the schematic. A small heating element was incorporated into the infrared thermometer to maintain its internal com-

ponents when it was operated at temperatures lower than -15 °C. Both the video camera and the infrared thermometer were

placed on adjustable stands, which allowed vertical and horizontal adjustments. In addition to the infrared thermometer, an-

other temperature sensor (PT-100) was placed inside the Plexiglas housing to monitor the thermal stability of the setup during

experiments.

The retention experiments were carried out inside the walk-in cold room of the laboratory at temperatures between -15 and

-28 °C. Silver iodide (AgI; Sigma Aldrich-99%) was used as the ice nucleating particle (INP) to initiate the freezing process.

We first characterized the INP at three different concentrations (0.2, 0.01, and 0.0003 g/L) at three different experimental tem-

peratures (-15, -20, and -28 °C). This provided the freezing curves of silver iodide at various temperatures and concentrations

(Fig. A2); more details can be found in Appendix A. These steps were a pre-requisite for retention experiments to infer the

correct drop freezing temperature ranges during our measurements.



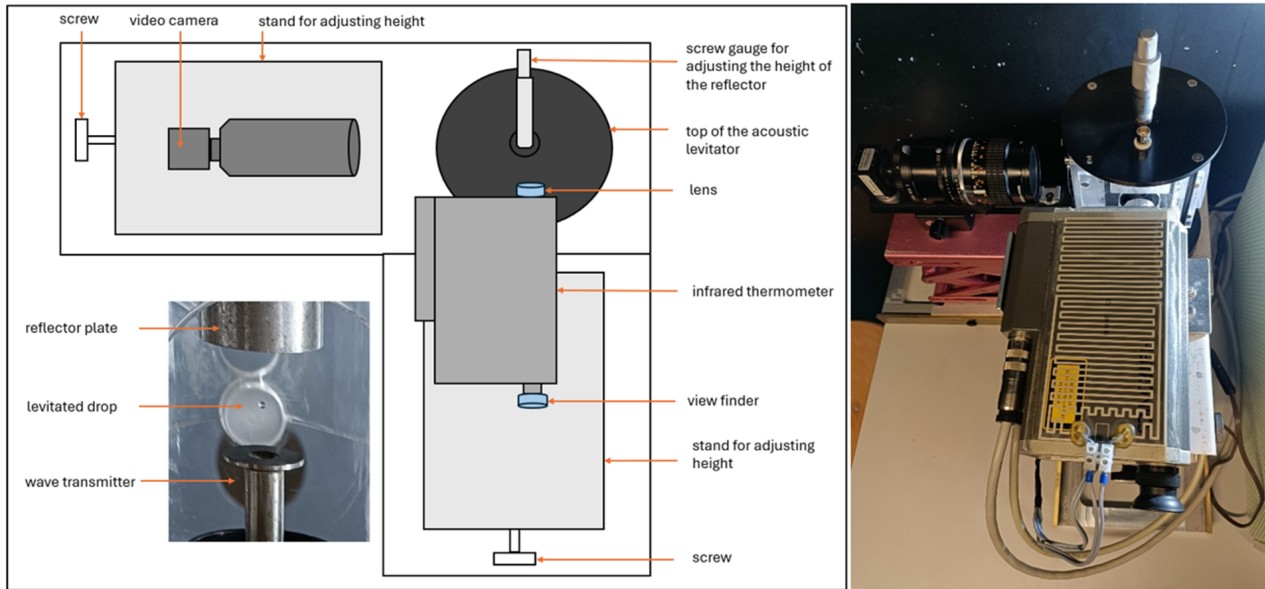

**Figure 1.** The Mainz Acoustic Levitator (M-AL) setup. Left: A schematic of the setup. Right: Setup in-situ.

.

## 2.2 Sampling procedure

In total, the retention of 4 single components and 3 binary mixtures were investigated. Nitric acid, formic acid, acetic acid, and 2-nitrophenol were measured as single components. Two sets of combinations were studied for the binary mixture of a strong and a weak acid, namely: nitric acid and acetic acid (mixture A1) as well as nitric acid and formic acid (mixture A2). Another set of binary mixtures was the combination of a small and a large molecule, due to their differences in molecular size and mobility. Here we investigated the mixture of formic acid and 2-nitrophenol (mixture B). The substances along with their purity labels are listed in Table 1.

Aqueous solutions of the investigated substances were prepared at an initial concentration of about 20 mg/L. These solutions were transferred to a syringe for injecting a single drop inside the M-AL. For each experiment 11 measurement points were recorded. Each measurement point consisted of a total of 10 frozen drops collected in a vial. The volume of one frozen drop was approximately 4.2 μL which makes the total volume for one measurement point being about 42 μL. These frozen drops were diluted 10 times in order to increase the injection volume for chemical analysis and filtered with a 2 μm pore size filter (Carl Roth GmbH).

Subsequent quantitative analysis was done using a DIONEX-ICS 1000 anion Ion Chromatography unit (IonPac AS9-HC column, 9 μm particle size, 4 × 250 mm dimension, Thermo Fisher Scientific Inc.) for nitric, formic and acetic acid. 2-nitrophenol and 2-nitrobenzioc acid were analysed with a high precision liquid chromatography (HPLC) unit (Hypersil GOLD column, 9 μm particle size, 150 × 2.1 mm dimension, Vanquish-Thermo Fisher Scientific Inc.).





**Table 1.** Substances and mixtures investigated in this study.

| Substance | Label/purity | Tracer | Concentration (mg/L) |
|---|---|---|---|
| Single components | | | |
| Nitric acid | Merck (65% w/w) | Sulphate[+] | 20 |
| Acetic acid | Riedel-de Haen (100%) | Nitrate [*] | 20 |
| Formic acid | Emsure (98-100%) | Nitrate [*] | 20 |
| 2-nitrophenol | Thermo Scientific (99%) | 2-nitrobenzoic acid[**] | 20 |
| Binary mixtures | | | |
| A1. Nitric acid and Acetic acid | – – | Sulphate[+] | 20 |
| A2. Nitric acid and Formic acid | – – | Sulphate[+] | 20 |
| B. Formic acid and 2-nitrophenol | – – | Nitrate[*] and 2-nitrobenzoic acid[**] | 20 |

Tracers: [+] Sulphate standard ($SO_4$): TraceCERT (99%), [*] Nitrate standard ($NO_3$): TraceCERT (99%), [**]2-nitrobenzoic acid: Thermo Scientific (95%)

Hydrochloric acid ($HCl$): Roth (37% w/w) and Sodium Hydroxide ($NaOH$): Merck (99%) were used to adjust the pH in the sensitivity studies.

The label/purity of the substances in binary mixtures are same as that of the single components.

For each of the investigated substances a concentration tracking tracer was added in order to track changes in mass concentration during the quantitative analyses. A tracer is a known chemical substance that is completely retained in ice, i.e. it has a retention coefficient of 1. The tracers used in this study were nitrate, sulphate, and 2-nitrobenzoic acid, which had a known retention of 1 from previous riming-retention studies (von Blohn et al., 2011; Borchers et al., 2024).

### 2.3    Calculation of retention coefficient

The retention during freezing was quantified by the retention coefficient $R$. It is the fraction of the chemical species that remains inside the frozen drops in the ice phase and the original solution in the liquid phase. The mathematical expression for calculating the initial retained fraction is given by

$$R_i = \frac{\frac{[substance]_{ice\ phase}}{[tracer]_{ice\ phase}}}{\frac{[substance]_{liquid\ phase}}{[tracer]_{liquid\ phase}}} \qquad (1)$$

In Eq. 1, the square brackets indicate the concentration of the investigated chemical species and the tracers and $R_i$ is the
retention coefficient without any correction for desorption. The numerator is the ratio of ice phase concentration of the measured species with their specific tracer, whereas the denominator is the ratio of liquid phase concentrations.

### 2.3.1    Correction for desorption

The freezing of the levitated drops is not an instantaneous process when injected into the acoustic trap. The drop is initially at a temperature higher than 0 °C. It then undergoes gradual supercooling until the freezing is initiated (Fig. A1). During this stage,
starting from injection of the drop into the acoustic field of the levitator and its subsequent progression to the supercooling





stage, the drop is exposed to external and internal forces until it is in equilibrium with its surroundings. Effects from the acoustic field potentially enhance ventilation while thermal stabilization can produce evaporation and desorption, leading to changes in aqueous concentration in the supercooled state. To account for all these effects, a correction parameter, called the desorption correction parameter $D$, was introduced:

$$D = \frac{\frac{[substance]_{supercooled\ phase}}{[tracer]_{supercooled\ phase}}}{\frac{[substance]_{liquid\ phase}}{[tracer]_{liquid\ phase}}} \tag{2}$$


To determine $D$, experiments were conducted under similar conditions as the retention experiments, with the exception of not adding any INP. In this case, the freezing process was not initiated and the liquid drop remained at a supercooled stage for a longer time. The drop was kept suspended for about 15 to 20 seconds, which is a typical time for the onset of freezing of the levitated drops under these experimental conditions (Fig. A1). Afterwards, the supercooled drops were instantly frozen inside

a liquid nitrogen bath, which has a temperature of about -197 °C (Scott, 1976; Jost et al., 2017). At such cold temperatures all substances inside the drops are retained during freezing. Quantitative analysis of these drops provided us with the concentration of the chemical substances in their supercooled stage and allowed the characterization of the desorption process.

The final retention coefficients $R$ of the investigated chemical substances were calculated as:

$$R = \frac{R_i}{D} \tag{3}$$

Colder temperatures would essentially slow down the reaction kinetics for desorption to be effective (Mitra and Hannemann, 1993; Seinfeld and Pandis, 2016). For experimental temperatures below -15 °C, desorption would play a negligible role. We applied the desorption corrections measured at -15 °C for substances measured at lower temperatures as well.

## 2.4 Sensitivity studies

Retention experiments with the investigated substances were also carried out at different pHs and temperatures. pH sensitivity
of the single components and the binary mixtures were studied at pH values of 3, 4, and 6/7. Hydrochloric acid (HCl) was used to lower the pH of the original solution and sodium hydroxide (NaOH) was used to increase the pH of the solution. The temperature sensitivity studies were performed at $-3.9 \pm 0.3°C$ and $-6.9 \pm 1.1°C$ drop freezing temperatures. These two different temperature ranges were evaluated from the INP freezing profiles (more details in Appendix A2). From the freezing profile obtained for experiments conducted at -15 °C cold room temperatures and 0.2 g/L AgI, the 50% frozen fraction
was found to be around $-3.9 \pm 0.3°C$, which was taken as the average drop freezing temperature under these experimental conditions (Fig. A2). Similarly, retention experiments were conducted at -23 °C cold room temperatures and AgI concentration of $0.008 \pm 0.001$g/L as the second experimental condition. The average drop freezing temperature for this second set of experimental conditions was obtained by extrapolating the freezing profile obtained at -20 °C cold room temperature and 0.1 unitg/L AgI (Fig A2), as the drop surface temperature cooling rates at -20°C and -23°C were practically identical (0.4





**Table 2.** Retention coefficients of the investigated substances at $-3.9 \pm 0.3°$C drop freezing temperature. The corresponding walk-in cold room temperatures (ambient temperature) was $-15 \pm 1°$C.

| Substance | Retention coefficient ($R$) |
|---|---|
| Single components | |
| Nitric acid | $1 \pm 0.03$ |
| Acetic acid | $0.88 \pm 0.12$ |
| Formic acid | $1.01 \pm 0.08$ |
| 2-nitrophenol | $0.90 \pm 0.05$ |
| Binary mixtures | |
| A. Mixture of a strong and a weak acid | |
| 1. Nitric acid and Acetic acid | Nitric : $0.97 \pm 0.06$ |
| | Acetic : $0.86 \pm 0.15$ |
| 2. Nitric acid and Formic acid | Nitric : $0.99 \pm 0.05$ |
| | Formic : $0.99 \pm 0.03$ |
| B. Mixture of a large and a small molecule | |
| Formic acid and 2-nitrophenol | Formic : $1 \pm 0.07$ |
| | 2-nitrophenol : $1.01 \pm 0.09$ |

°C/s). The 50% frozen fraction at -23°C was found to be $-6.9 \pm 1.1°$C. The two temperature ranges were selected to compare the temperature sensitivity in earlier experiments concerning retention coefficients for cloud sized droplets (von Blohn et al., 2011, 2013; Jost et al., 2017; Borchers et al., 2024). The average size of the droplets was $21.5 \pm 8.5$µm in the above mentioned studies involving riming-retention. In the present freezing-retention study with large levitated drops, the average drop sizes were $2.0 \pm 0.1$mm.

## 3   Results and Discussions

### 3.1   Retention coefficient

The final retention coefficients for single components and binary mixtures are shown in Table 2. It can be seen that most of the substances measured as single components were completely retained in the ice phase. The exceptions were acetic acid and 2-nitrophenol, which were found to have retention coefficients of 0.88 and 0.90, respectively. However, for acetic acid as a single component, the standard deviation was much larger ($\pm$ 0.12) compared to the other single component substances. Thus, acetic acid could also be completely retained during freezing. The standard deviation of 2-nitrophenol was smaller compared to acetic acid and it was the least retained substance (0.85 to 0.95) of investigated single components.

Brand (2014) studied the retention of large drops (2.67 mm and 7.25 mm spherical equivalent diameter) by freezing them on a Teflon coated pallet and also reported high retention coefficients (close to 1). For example, for drop sizes of 2.67 mm (i.e.



10 μL drop volume), formic acid showed a retention coefficient of 0.94 ± 0.04. However, in our study contact-free immersion freezing (Diehl et al., 2014; Szakáll et al., 2021) was employed with which a more realistic scenario was realized to initiate freezing as in Brand (2014). Nevertheless, the measured retention coefficients in the present freezing-retention study and in Brand (2014) indicate near complete retention for the large rain sized drops.

Comparing our present results from freezing-retention experiments with previous riming-retention studies (von Blohn et al.,
2011, 2013; Jost et al., 2017; Borchers et al., 2024), one can observe a deviation from their findings. Retention coefficients measured for cloud sized droplets during riming-retention experiments show a sigmoidal dependency on the solubility and dissociative properties of the individual substances (i.e. their effective Henry's law constant $H^*$). Our present experiments do not reveal these observed dependencies for the large rain sized drops. For instance, 2-nitrophenol (as single component in Table 2) having the least $H^*$ among the investigated substances, was highly retained inside a freezing raindrop indicated by a
retention coefficient of 0.9. However in the case of riming-retention, 2-nitrophenol showed a retention coefficient of 0.12 at pH 4 and 0.27 at pH 5.6 (Borchers et al., 2024). Further discussion comparing the results from riming-retention of cloud droplets and freezing-retention of raindrops from this study is provided in Section 3.4.

In the binary mixture experiments, in which we combined a strong and a weak acid (A1 and A2 in Table 2), nitric acid was the stronger acid with a pKa value of -1.3 (Haynes, 2016). Acetic acid and formic acid, having pKa values of 4.76 and 3.77
respectively, were the weaker acids compared to nitric acid. The results shown in Table 2 indicate that binary mixtures do not seem to alter the retention coefficients of their individual species for the combination of a strong and a weak acid.

Mixture B had the combination of a small and a large compound. There the average retention coefficient of 2-nitrophenol in a mixture with formic acid was observed to have increased slightly as compared to its retention as a single component. As a binary mixture component, both 2-nitrophenol and formic acid are completely retained during freezing.

**3.2 pH Sensitivity**

Retention coefficients of the single components were each measured at three different pH values. As a strong acid, nitric acid completely dissociates and is therefore assumed to be completely retained. Hence sensitivity studies for nitric acid were not done.

The original aqueous solutions at a concentration of 20 mg/L had pH values around 4.2 and 4.4 for each of the three
substances shown in Fig. 2a. Acetic acid (green marker) and formic acid (blue marker) did not show any apparent dependency on pH. An argument could be made for acetic acid as its retention coefficient seems to increase with increasing pH, however, the standard deviation for each 11 sets of measurements at the three different pH values was quite large. The retention coefficients for acetic acid were 0.81, 0.88, and 1.05 for pH values of 3.1, 4.2, and 7.0, respectively, while their corresponding standard deviations were 0.18, 0.12, and 0.2. For 2-nitrophenol (red marker) an increase in retention (1.05) can be seen at pH 6. At lower
pHs of 3.2 and 4.4, retention coefficients were about 0.90 for both cases. From Fig. 2a, one can infer a slight dependency on pH for 2-nitrophenol, and almost none for acetic acid and formic acid.





**Figure 2.** pH sensitivity of the retention coefficient of (a) single components, and (b) binary mixtures.

The pH sensitivities for the binary mixtures are shown in Fig. 2b. Mixture A1 was omitted due to the larger standard deviation for acetic acid compared to formic acid as a single component. In mixture A2, both substances were retained completely. The same was found for mixture B. As shown in Fig. 2b, none of mixtures show any sensitivity to changes in pH.

pH of the solutions were altered by adding $HCl$ and $NaOH$, which could also interact with the investigated substances and dissociate them into their ionic form. In this case the overall concentration of the investigated substances could be lowered. A lower concentration of the substances, in turn, might not have enough partial pressure in the liquid phase (inside the levitated drops before freezing) for them to be expelled from the drop when it freezes. However, after the addition of $HCl$ or $NaOH$, the initial concentration of $20 \ \mathrm{mg/L}$ of the investigated substances did not change much. After addition, the lowest measured initial liquid phase concentration was $17.8 \ \mathrm{mg/L}$ (11% decrease). The total number of moles of the dissolved substances had the same order of magnitude as their initial liquid phase concentration in the solution. Thus, any significant source of biases towards a higher retention coefficient due to addition of $HCl$ or $NaOH$ can be neglected in our measurements.





**Figure 3.** Temperature sensitivity of the retention coefficient of (a) single components, and (b) binary mixtures.

### 3.3 Temperature Sensitivity

The temperature sensitivities for the single components are shown in Fig. 3a. Acetic acid (green marker) showed a higher
retention coefficient at the lower temperature with large standard deviations of the measurements at both temperatures. At
-6.9°C the retention coefficient for acetic acid was 1.14±0.24 and at -3.9°C it was 0.88±0.12. Formic acid (blue marker) did
not show any variation in retention coefficient with changes in the drop freezing temperatures and was completely retained at
both temperatures. 2-nitrophenol (red marker) also had a higher retention coefficient at the colder temperature (1.06±0.05) as
compared to the warmer temperature (0.90±0.08). The retention coefficients for both acetic acid and 2-nitrophenol appeared
to have a weak dependency on temperature and were completely retained at (-6.9°C) along with formic acid, which had no
dependency and was completely retained at both temperatures. In the atmosphere, freezing is initiated at lower temperatures
than our experimental temperatures here, indicating towards complete retention of the investigated species.




Unlike the single components, the binary mixtures did not show any temperature dependency as seen in Fig. 3b. Both sets of binary mixtures were fully retained at $-3.9 \pm 0.3°\mathrm{C}$. At the colder temperature, the retention coefficients did not change and the mixtures were completely retained.

### 3.4 Relation with effective Henry's law coefficient

Retention coefficients of substances are strongly dependent on chemical properties such as aqueous diffusion, gaseous diffusion, interfacial mass transport, solubility and dissociation. Among them, solubility and dissociative effects characterized by effective Henry's law constant $H^*$ were reported to be the dominant ones. Stuart and Jacobson (2003) and Jost et al. (2017) showed this relationship between the retention coefficient and $H^*$, where they stated that substances with $H^*$ greater than $10^7$ are completely retained. Substances with $H^*$ lower than $10^4$ are less likely to be retained or more likely expelled from the drop during riming-retention. Retention coefficients of all other substances with $H^*$ values between these ranges followed a sigmoid shape (see Borchers et al., 2024, Figure 7).

The relation between effective Henry's law coefficient and retention coefficient for retention-riming was modeled by the following equation:

$$R_{H^*} = \left[1 + \left(\frac{a}{H^*}\right)^b\right]^{-1} \tag{4}$$

where the parameters $a = (2.41 \pm 1.06) \times 10^4$ and $b = 0.27 \pm 0.04$, respectively (Borchers et al., 2024).

Figure 4 shows the relation between $H^*$ and $R$. The gray markers are from previous studies (von Blohn et al., 2011; Jost et al., 2017; Borchers et al., 2024) for riming-retention. The coloured markers are from the present study utilizing freezing-retention.

It is apparent from Fig. 4, that nitric acid with an $H^*$ of $10^{11}$ was completely retained. Formic acid was completely retained, too, which is in contrast to previous measurements from riming-retention studies in which it showed a lower retention coefficient (0.76). No definitive conclusion regarding changes in its measured retention coefficient can be made for acetic acid (0.88 for single component), due to large standard deviation and the overlap between the single components and the binary mixture measurements. Conversely, the riming-retention of acetic acid was much lower (0.6). 2-nitrophenol showed a much higher retention coefficient for large drops (0.9 and above) compared to its retention for small μm sized droplets (0.27 at pH 5.6; Borchers et al. 2024). Considering its low $H^*$ ($10^3$), one would expect the retention coefficient of 2-nitrophenol as a single component to be lower than 0.9, which was not the case here. In the mixture with formic acid, 2-nitrophenol was also completely retained. Specifically, Fig. 4 demonstrates that our results from freezing-retention deviate from the sigmoidal relationship between retention coefficients and $H^*$ unlike the previous experimental studies involving riming-retention. This result is also seen in the conclusions of Part II of this publication series, where the retentions for ambient water soluble organic compounds of over 450 species were also investigated.





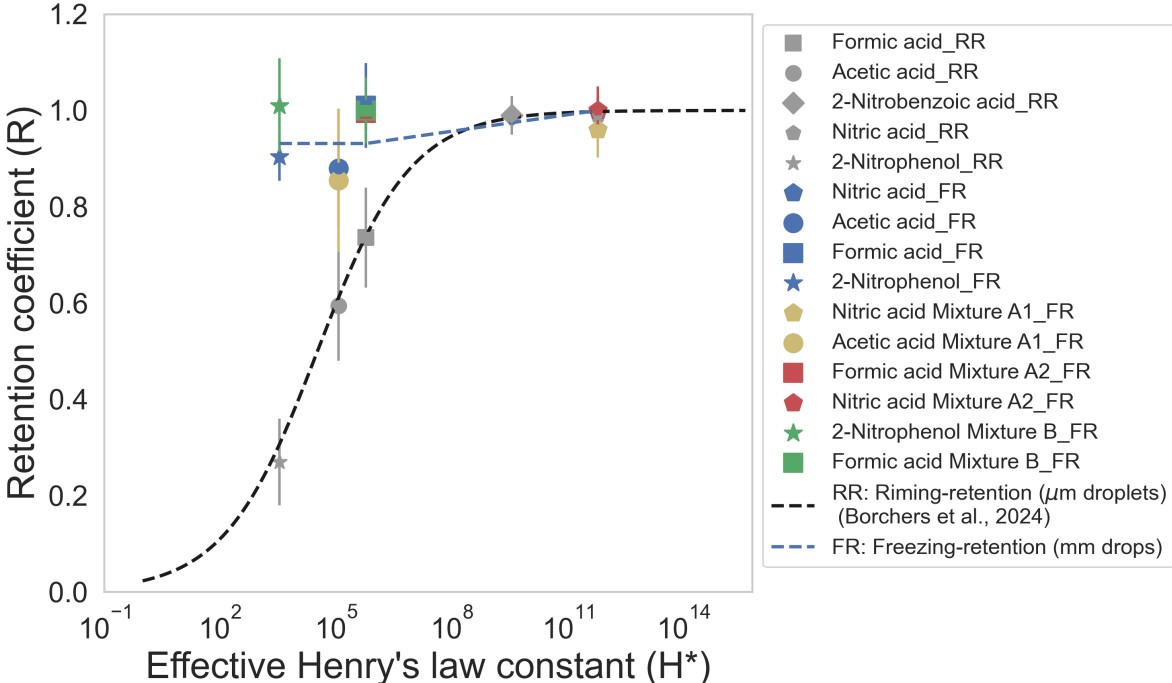

**Figure 4.** Retention coefficient ($R$) as a function of effective Henry's law coefficient ($H^*$). Grey markers: from riming-retention (RR) of small droplets (von Blohn et al., 2011; Jost et al., 2017; Borchers et al., 2024). Colored markers: from freezing-retention (FR, present study), with drop freezing temperature of $-3.9 \pm 0.3°$C. Blue markers: single components. Yellow (mixture A1), red (mixture A2) and green (mixture B) markers: binary mixtures.

### 3.5 Retention indicator analysis

Another method to analyze retention is from the point of view of mass and heat transfer considerations, such as the mass
expulsion and freezing timescales as suggested by Stuart and Jacobson (2003, 2004) and Jost et al. (2017). Retention indicator ($RI$) is introduced that is the ratio of total mass expulsion time ($T_{exp}$) to the freezing time ($T_{frz}$) as shown in Eq. (5). Table 3 shows the calculated timescales for the retention indicator of the single components investigated in this study.

$$RI = \frac{T_{exp}}{T_{frz}} \qquad (5)$$

$$T_{exp} = T_g + T_{aq} + T_i \qquad (6)$$

where,    $T_g = \frac{a^2 H^*}{3 D_g f}$;    $T_{aq} = \frac{a^2}{D_{aq}}$;    $T_i = \frac{4 a H^*}{3 v \alpha}$



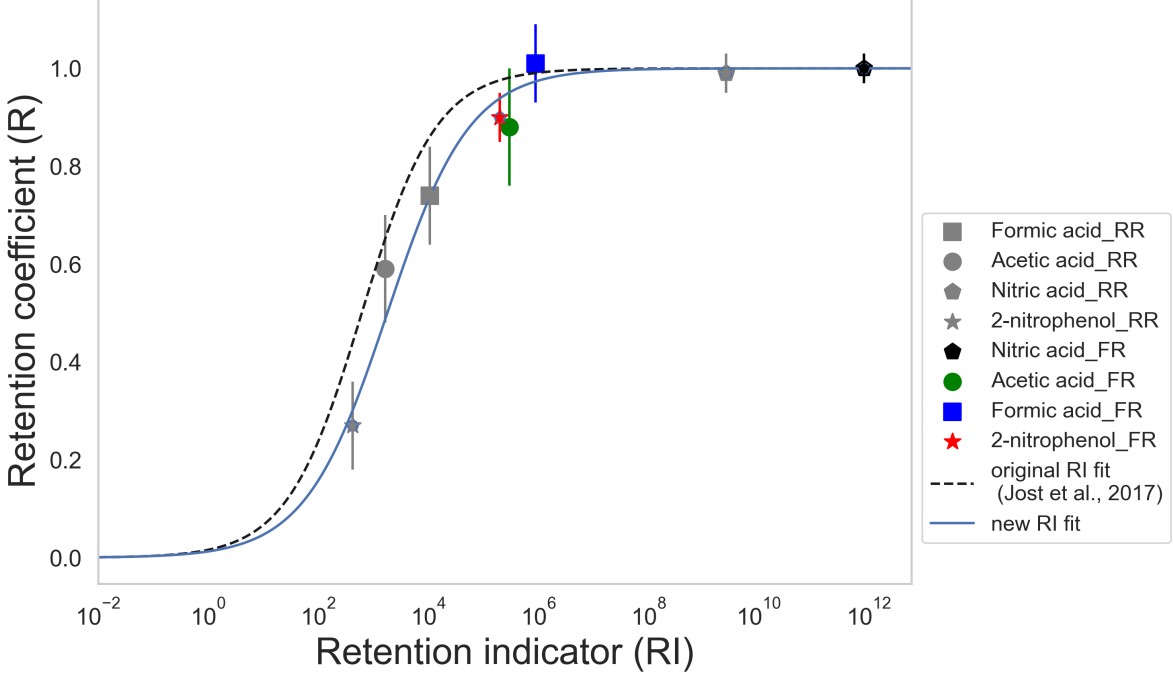

**Figure 5.** Retention coefficient of the substances investigated as single components as a function of the empirical retention indicator. Gray markers: from riming-retention (RR), coloured markers: from freezing-retention (FR). Dashed line: original retention indicator fit parameters from Jost et al. (2017), solid line: updated fit from current study.

The total solute mass expulsion time $T_{exp}$ is the sum of aqueous phase mass expulsion time $T_{aq}$, gaseous phase mass expulsion time $T_g$, and interfacial mass transfer expulsion time $T_i$. In Eq. (6) $T_g$ accounts for the gaseous diffusivity $D_g$, where $a$ is the radius of the drop, $H^*$ is the effective Henry's law coefficient and $f$ is the ventilation coefficient ($f$= 5.6; Szakáll et al. 2021). $T_{aq}$ accounts for the aqueous diffusivity $D_{aq}$ of the substance. $T_i$ takes into consideration for the mass accommodation coefficient $\alpha$ and the thermal velocity of the chemical in air, $v$ .

A fourth timescale involving the aqueous phase kinetics was also introduced by Jost et al. (2017). This timescale is specifically important for substances such as ammonia and formaldehyde since they react with atmospheric carbon dioxide and are affected by dehydration of methanediol, respectively (Jost et al., 2017). For substances investigated in this study, the aqueous phase kinetics and reactions are negligible, so this timescale was not considered. The experimental temperatures, pH values, initial concentrations and $H^*$ are also listed in Table 3 for reference. The freezing time $T_{frz}$ was derived experimentally via high speed camera (Motion Pro Y3M; pixel size: 12 × 12 μm; resolution: 1024 × 1280 pixels) at 600 frames per second, as shown in Fig. B1. The time from the initiation of freezing to the complete formation of an ice shell around the levitated drop was approximately 4.8 ms. This value was taken as the freezing time for the retention indicator calculation. It is also clearly evident in Table 3 that $T_{frz}$ is several orders of magnitude smaller compared to $T_{exp}$. Gas phase expulsion time $T_g$ appears to





**Table 3.** List of parameters used for retention indicator calculation.

| Parameters | Nitric acid | Acetic acid | Formic acid | 2-nitrophenol | Comments |
|---|---|---|---|---|---|
| $^1D_{aq}$ | $2.25\times10^{-5}$ | $1.29\times10^{-5}$ | $1.63\times10^{-5}$ | $1.07\times10^{-5}$ | Aqueous diffusivity ($cm^2/s$) |
| $^1D_g$ | 0.12 | 0.12 | 0.14 | 0.07 | Gaseous diffusivity ($cm^2/s$) |
| pH | 4.1 | 4.2 | 4.2 | 4.4 | Experimental pH values |
| $^2H^*$ | $7.56\times10^{11}$ | $1.28\times10^{5}$ | $8.31\times10^{5}$ | $3.50\times10^{3}$ | Effective Henry's law constant |
| $^3\alpha$ | 0.06 | 0.07 | 0.05 | 0.01 | Mass accomodation coefficient |
| T | -3.9 | -3.9 | -3.9 | -3.9 | Temperature ($^\circ$C) |
| C | 20 | 20 | 20 | 20 | Concentration (mg/L) |
| $T_g$ | $3.88\times10^{9}$ | $6.37\times10^{2}$ | $3.55\times10^{3}$ | $2.81\times10^{1}$ | Gas phase expulsion time (s) |
| $T_i$ | $2.11\times10^{7}$ | $7.78\times10^{0}$ | $6.36\times10^{1}$ | $2.19\times10^{0}$ | Interfacial expulsion time (s) |
| $T_{aq}$ | $6.81\times10^{2}$ | $7.72\times10^{2}$ | $6.13\times10^{2}$ | $9.28\times10^{2}$ | Aqueous phase expulsion time (s) |
| $T_{exp}$ | $3.90\times10^{9}$ | $1.42\times10^{3}$ | $4.23\times10^{3}$ | $9.59\times10^{2}$ | Total expulsion time (s) |
| $T_{frz}$ | $4.80\times10^{-3}$ | $4.80\times10^{-3}$ | $4.80\times10^{-3}$ | $4.80\times10^{-3}$ | Ice shell formation time (s) |
| RI | $8.13\times10^{11}$ | $2.95\times10^{5}$ | $8.80\times10^{5}$ | $2.00\times10^{5}$ | Retention indicator |
| R | $1.00 \pm 0.03$ | $0.88 \pm 0.12$ | $1.01 \pm 0.08$ | $0.90 \pm 0.05$ | Retention coefficient |
| Controlling parameter | $T_g$ | $T_{aq}$ | $T_g$ | $T_{aq}$ | – |

$^1$The diffusivities in water $D_{aq}$ and in air $D_g$ calculated at 273K (Thibodeaux and Mackay, 2010), $^2$Effective Henry's law constant calculated at 273K and at their corresponding pH (Tremp et al., 1993; Johnson et al., 1996; Warneck and Williams, 2012), $^3$ The mass accommodation coefficient at 273K (Ervens et al., 2003; Davidovits et al., 2006).

be the controlling factor contributing to the total high $T_{exp}$ for nitric and formic acid, and aqueous phase expulsion timescale $T_{aq}$ for acetic acid and 2-nitrophenol. In Jost et al. (2017) the parameterization relating $RI$ and retention coefficient is given as

$$R_{RI} = \left[1 + \left(\frac{c}{RI}\right)^d\right]^{-1} \qquad\qquad (7)$$

Equation 7 is depicted in Fig. 5, where the original parameters taken from Jost et al. (2017) are $c_1 = 618\pm71$ and $d_1 = 0.64\pm0.06$ (black dashed line, Fig. 5). From our study, an updated fit is provided with $c_2 = 1800\pm95$ and $d_2 = 0.58\pm0.07$ (blue

solid line, Fig. 5).

Figure 5 shows the variation of retention coefficients with $RI$. In contrast to Fig. 4, both the riming-retention and freezing-retention measurements fit well with the parameterization given in Eq. 7. This analysis corroborates our experimental results for mm sized raindrops with µm sized cloud droplets. These results can be categorized with timescale analysis and follow a similar relation with both previous experimental (Jost et al., 2017) and theoretical (Stuart and Jacobson, 2003, 2004) studies.





 ## 3.6 Physical parameters

Our study shows that retention is dependent on the size of the droplets which needs to be considered when modeling the mass flux of trace substances with numerical models. An aspect of the importance of the physical parameters is surface area to volume ratio. The rain sized drops in this study have a surface area to volume ratio of $3 \times 10^3$ m$^{-1}$. The cloud sized droplets in earlier retention-riming studies have a surface area to volume ratio of about $2 \times 10^7$ m$^{-1}$. Thus, this ratio is approximately 285 4 orders of magnitude higher for the cloud droplets compared to the rain drops. As such, the dissolved substances in raindrops would have more diffusional volume and smaller surface area. Additionally, low surface area to volume ratio for the case of the rain drops is an indicator of lower overall desorption as well (Jost, 2017).

Another physical parameter influencing retention is the ventilation coefficient. It describes the enhanced heat and mass transfer around hydrometers in an airflow. For the riming-retention studies, substances measured inside a wind tunnel (µm 290 sized droplets) had ventilation coefficients of about 30 to 32 (Jost et al., 2017, Table 4). In contrast, the ventilation coefficient in the acoustic levitator for the 2 mm diameter drops was about 5.6 (Szakáll et al., 2021). As such, a smaller ventilation coefficient would incur less transfer of mass and heat for the 2 mm raindrops as compared to the retention measurements for µm sized droplets. This could be seen as an important physical parameter aiding higher expulsion times and, consequently, higher retention coefficients as seen in the RI analysis. In a real atmospheric scenario, 2 mm drops falling at their terminal 295 velocity have a ventilation coefficient of about 15 (Pruppacher and Klett, 2010). A higher ventilation coefficient would increase the mass transfer and thereby decrease the expulsion timescale. However, the ventilation coefficients of heat and mass transfer is almost the same. Therefore an increase in mass transfer would also imply a faster freezing time.

## 4 Conclusions

Our results show higher retention coefficients close to 1 for mm sized raindrops for similar substances from previously studied 300 retention coefficients (von Blohn et al., 2011; Jost et al., 2017; Borchers et al., 2024) in µm sized cloud droplets. It is important to note that in addition to the differences in droplet size, the freezing pathways were also different. The previous studies utilized the riming-retention mechanism while in the present work we incorporated a contact-free freezing-retention mechanism.

Substances studied as single components show very little sensitivity (for 2-nitrophenol and acetic acid) with changes in either pH or temperatures. Formic acid as a single component is not sensitive to changes in pH or temperatures. Binary mixtures also 305 do not show any sensitivity for changes in pH and freezing temperature.

We conclude that for rain sized drops (mm and above), most of the chemical species are completely retained during freezing. This can be interpreted as the physical parameters -— such as drop size and ice shell formation -— dominating the chemical properties concerning retention influences. After an ice shell is formed around a drop during the initiation of freezing, it is significantly more difficult for the dissolved species to be expelled from the drop, thus leading to higher mass expulsion 310 timescales.

Our retention indicator analysis shows that the shorter freezing and longer expulsion timescales (a minimum of 5 orders of magnitude higher) lead to higher retention for the investigated species. This indicates that during the freezing of mm sized





raindrops all dissolved trace gases may be removed entirely by precipitation in deep convective clouds or transported within the ice phase into the UT where it can be released upon sublimation.

We derived new parameterizations for the retention indicator to include large mm sized raindrops, and thus, updated the previously obtained ones that only considered µm sized cloud droplets (Jost et al., 2017). This result is beneficial in terms of computational expense for the chemistry coupled atmospheric and earth system modelling as modelling freezing raindrops would not require much additional computational resources.

   Our experiments were conducted with single components and binary mixtures but in the real atmosphere, air is mixed with
numerous complex trace gases that are in constant turbulent motion. Our current database does not have many substances with $H^*$ values lower than $10^3$, and such substances might behave differently during freezing. Future retention experiments that sample for trace gas at the ground level and at different vertical profiles would improve our understanding of the underlying micro-physical and chemical processes within convective systems. Our experiments also indicate that it is critical to further investigate the ice shell formation process during the freezing of raindrops.

Future studies should investigate how these and similar organic compounds behave when they are in the real atmosphere. In Part II we investigate the retention of a complex mixture of organic compounds sampled from Beijing urban aerosols through the same experimental setup with high resolution mass spectrometry analysis.

*Data availability.* The data supporting this study are available at the repository Gautam and coauthors (2024). Additional data (if required) for this study are available upon request from the corresponding authors.

**Appendix A:  Characterization of INP**

**A1   Freezing profile of levitated drops**

To characterize the INP (AgI) we levitated drops and recorded their drop surface temperature as they froze, at three pairs of different concentrations and cold room temperatures: 0.2, 0.01, and 0.0003 g/L, and corresponding cold room temperatures of -15, -20, and -28°C. For both combinations of concentration and temperature, the freezing profiles of about 50 drops were
recorded. The crucial information derived from these three sets of measurements was obtaining the freezing profiles of the levitated drops during their freezing. Figure A1 shows a typical drop freezing profile as the temporal evolution of the drop surface temperature. The drop when injected to the nodes of the standing wave, had initially a temperature higher than 0°C. The warm drop underwent gradual and uniform cooling and reached a supercooled stage (0 to 20 seconds). The supercooled stage continued until nucleation was initiated, where the rapid crystal growth started (about 25 seconds) and drop surface
temperature rose sharply to about 0°C. The rapid crystal growth can be interpreted as adiabatic freezing and the corresponding temperature was taken as the freezing temperature of the drop. At this temperature, the supercooled drop entered an ice-water equilibrium, visible as the flattened section in Fig. A1 (30 to 80 seconds). During this stage, transfer of latent heat took place that can be interpreted as the diabatic freezing of the levitated drop. The supercooled drop then underwent a phase transition





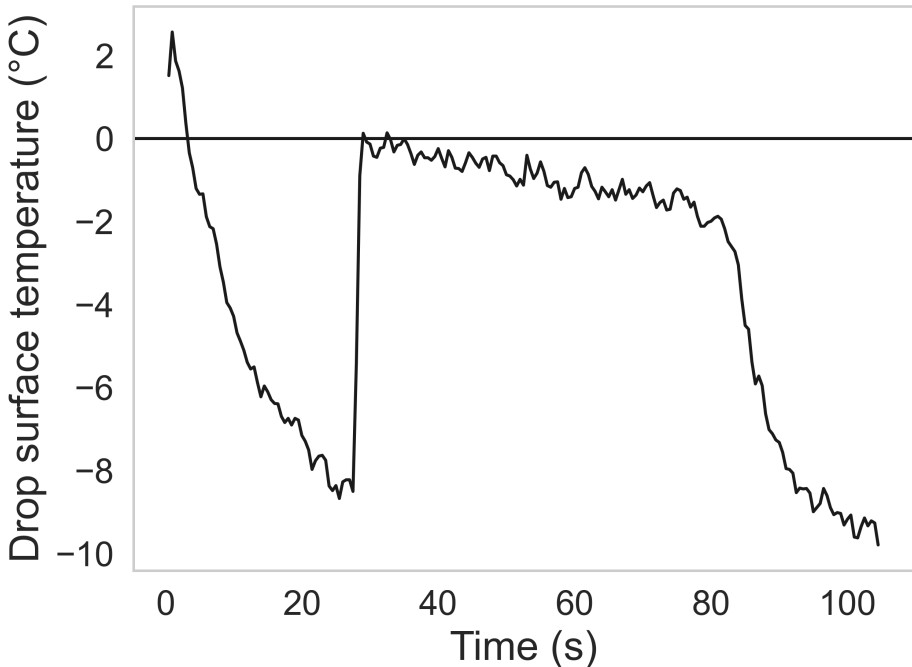

**Figure A1.** Evolution of drop surface temperature during its freezing as measured by the infrared thermometer.

from liquid to solid state. Finally, the drop surface temperature cooled down to ambient temperature, reaching a steady state
(100 seconds) once it was completely frozen.

## A2 Frozen fraction

Within the range of the sample size of 50 drops for each set of frozen fraction measurements, the precise drop freezing
temperatures varied. We grouped the recorded drop freezing temperatures in bins with a width of 0.5 °C. Corresponding to
each bin, the number of frozen drops at each interval were grouped. A cumulative distribution was formed with the grouped
bins. As commonly used in ice nucleation studies, frozen fraction or $f_{ice}$ was determined, which is calculated as the fraction
of total drops that were frozen at a particular temperature (more details in Szakáll et al., 2021). The temperature at which $f_{ice}$
was 50% was taken as the 'average drop freezing temperature' for each set of concentration and cold room temperature pair.

The frozen fractions for each set of measurements are shown in Fig. A2. The average drop freezing temperature was -3.9°C
for AgI concentration of 0.2 g/L and cold room temperature of -15°C. For the combination of 0.01 g/L and -20 °C, the
average drop freezing temperature was -6.7 °C and -8.9 °C for the combination of 0.0003 g/L and -28°C. We conducted our
retention measurements at a cold room temperature of -23°C. To obtain the freezing profile at this temperature, we refitted the
freezing profile obtained for -20°C using the following equation:



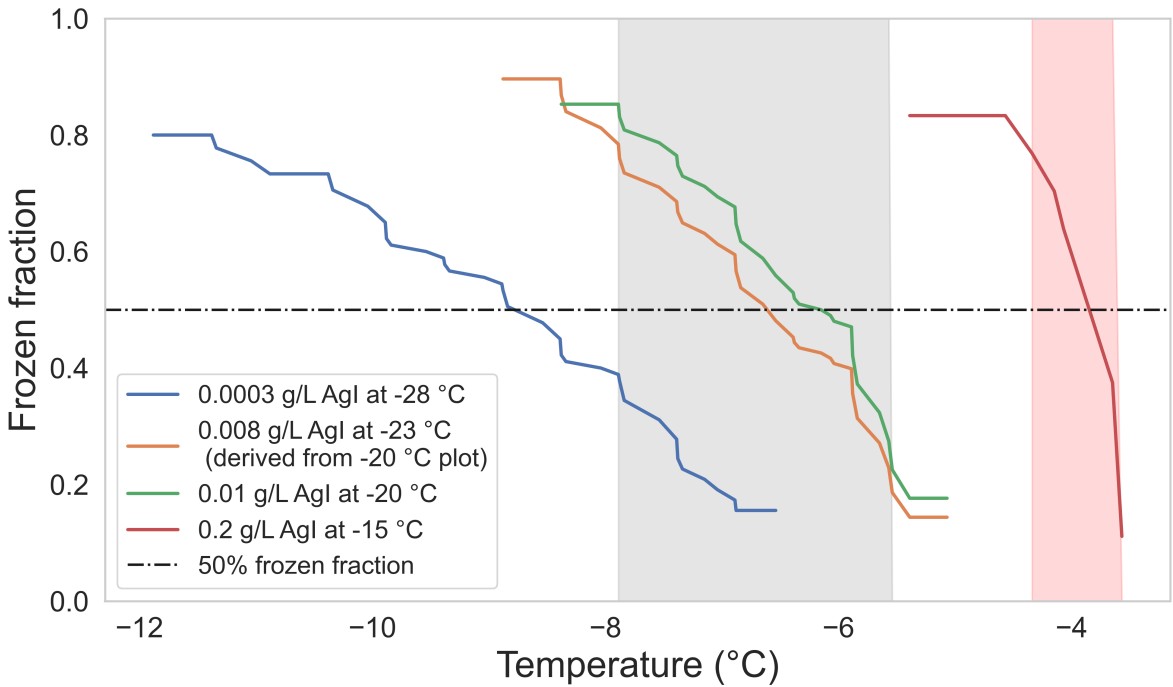

**Figure A2.** Frozen fraction at different ambient temperatures and concentrations of AgI. Shaded regions mark the two selected temperature ranges for retention measurements. The shaded regions lie within the interval where the frozen fraction is in between 0.8 and 0.2.

$$f_{ice\_23} = 1 - \exp \frac{c_{23} * \ln(1 - f_{ice\_20})}{c_{20}} \tag{A1}$$

where $f_{ice\_23}$ is the desired frozen fraction distribution at -23°C. $c_{20}$ and $c_{23}$ are the INP concentrations at the two different temperatures of -20°C and -23°C, respectively. $f_{ice\_20}$ is the experimentally derived frozen fraction at -20°C. The cooling rate of the drop surface temperature was practically identical at these two cold room temperatures. Equation A1 is adopted from the relation between ice nucleation active sites ($n_s$) and $f_{ice}$ and at a particular INP concentration and temperature (see Szakáll et al., 2021, Eq 5).

We selected the interval where frozen fraction lies between 20% to 80% as the temperature deviation during our retention experiments. Shaded regions in Fig. A2 show this temperature deviation for experiments done at -15°C and -23°C cold room temperatures. The average drop freezing temperatures (frozen fraction at 50%) in these two cases were $-3.9 \pm 0.3°C$ (red-shaded region) and $-6.9 \pm 1.1°C$ (gray shaded region).



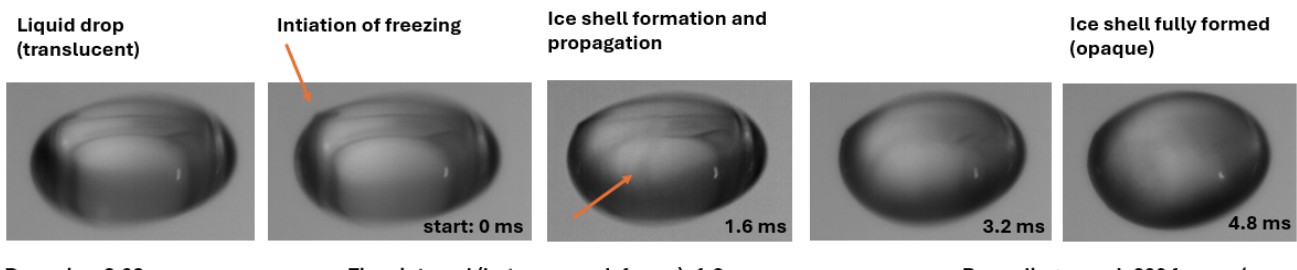

**Figure B1.** Consecutive frames showing the formation of ice-shell, recorded with a high speed camera at 600 frames per second, and at a cold room temperature of -15 °C. In the liquid phase (leftmost image), the drop is seen as translucent, which gradually turns opaque as the ice shell is formed (rightmost image).

## Appendix B: Ice shell formation during freezing

The investigation of the drop freezing mechanism in the acoustic levitator led to the realization of the ice shell formation. During the rapid crystal growth stage within the first 25 s, as discussed in Appendix A1, Fig. A1, an ice shell formed around the supercooled drop within milliseconds (Fig. B1). After the formation of the shell, freezing inside the drop proceeded gradually until it was completely frozen. The shell formation process was recorded with a high speed camera setup at 600 frames per second, and at a cold room temperature of -15 °C.

This observation validates the higher retention coefficients of the substances measured during our freezing-retention experiments, as compared to the previously measured substances involving riming-retention. The ice shell inhibited the expulsion of the dissolved chemical substances from the drop. The expulsion timescale as discussed and calculated in Section 3.5 was several orders of magnitude higher than the freezing time scale of 4.8 milliseconds (Fig. B1). This led to a higher value of the retention indicator, even for more volatile substances such as 2-nitrophenol, which had the lowest effective Henry's law

constant among the investigated substances (Figure 5 and Table 3).

*Author contributions.* MG, MZ, AT, JS, SM participated in designing the experiments; MG, MH performed the experiments; MG, MH, JS conducted analytical measurements, MG analysed the data and wrote the manuscript draft; AT, JS, MH, SB, KD, MZ reviewed and edited the manuscript

*Competing interests.* The authors declare no competing interests.



*Acknowledgements.* This work was funded by the Deutsche Forschungsgemeinschaft (DFG, German Research Foundation) – TRR 301 – Project-ID 428312742.



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
