# Peer review of "Retention During Freezing of Raindrops, Part I: Investigation of Single and Binary Mixtures of Nitric, Formic and Acetic Acids and 2-Nitrophenol"

_EGUsphere, 2024_

## Referee Comment (RC2)

Referee comment by Gabor Vali on "Retention During Freezing of Raindrops, Part I: Investigation of Single and Binary Mixtures" by Gautam et al.

   This manuscript extends to raindrops the work previously done by the authors and their colleagues on retention coefficients for cloud droplets. Acoustic levitation in a cold room and chemical analyses before and after freezing constitute the essence of the experiments. The levitation system and the use of infrared thermometry avoid the need for contact with any support. This is a near ideal arrangement. The paper present a good description of the experiments and sound analyses of the retention coefficients. The paper is well constructed and well written (with a few odd phrasing). The topic is quite appropriate for ACP.

   This reviewer has not been involved for decades with the topic of retention of foreign material from ice growth and will only address in these comments the physical aspects of the experiments, how to interpret the results, and to what extent the results apply to processes in atmospheric clouds.

   Two features of the experimental approach are the focus of these comments: the large difference in temperature between the drop and the surrounding air, and the near-absence of ventilation.

   The 2-mm diameter raindrops used in the experiments are up to 100 times larger than the cloud droplets used in the previous experiments and thus have about 100 times larger volume to surface ratios. That would lead one to anticipate significantly slower freezing and, consequently, larger rejection of foreign substances as ice forms. The results here presented show the opposite. For two of the substance involved in both experiments (formic and acetic acids), values near 0.7 were obtained in the riming experiments and near 1.0 for the raindrops. The authors' chief argument for this is that the larger drops in free air had an ice shell form on their outside trapping most of foreign substances.

   The formation of the ice shell after nucleation is well documented in the paper. It is also what one would expect for an isolated drop with the air temperature considerably lower than the drop temperature even before nucleation and pronouncedly so during the freezing of the drop when the surface temperature rises to near 0°C (Fig. A1). In contrast, in the atmosphere, the temperature of the drop would be close to the air temperature before freezing. It would also have asymmetric heat transfer when nucleation and initial ice formation leads to latent heat release within the drop. The resulting surface temperature and the formation of ice within the drop will be influenced by the asymmetry and by the rate of heat transfer to the environment. Theoretical analyses of the problem have been made with respect to hailstone formation and growth (e.g. List, 2014). These analyses also consider evaporation from the drop surface and collection of cloud droplets, but do not treat explicitly how ice forms inside the drop. For the current discussion, more relevant are the many experiments, and drops caught in clouds, that demonstrate that frozen drops often have protuberances and other deformation on their surfaces. Cracks in the ice shell may lead to the expulsion of liquid to the surface and perhaps to the air. The theory of ice multiplication in clouds by

splintering is based on those observations (Field et al, 2017; Lauber et al., 2018).

   The potential for cracks in the ice shell may also have to considered for the experiments described in the paper. Internal pressure rises as the drop freezes and is likely to produce cracks in the ice shell (e.g. Korelev and Leisner, 2020; Kleinheins et al., 2021; references herein). Because of the low air temperature in the experiments, any excluded water is likely to freeze onto the surface quite rapidly. This would slow internal freezing. The cited papers describe work with water without added substances. Dissolved gases or ions may modify the freezing behavior.

   Most of the foregoing work was done with droplets of hundreds of micrometer in diameter, not far but still below the size of 2 mm involved in the current experiments. That discrepancy and the complex nature of the phenomenon make any extrapolation difficult and it is even more speculative how all of the above influence retention of foreign substances. In that light, it is a welcome development to have the results presented in this paper. However, it is clear that more work is needed and that the authors of this paper should express their views on the matter in the manuscript.

   Another dimension of the problem is how the high retention found in this work might be envisaged on the molecular scale. Some discussion of the results of molecular simulations of crystal growth may help readers' understanding of the results.

   Section 3.5 deals with the retention indicator defined by the relative timescales of mass expulsion and that of freezing. It is unclear if this measure is intended to describe and idealized freezing front or is applied to specific geometries, spheres in this case. Perhaps the authors can illuminate this by justifying their choose of the parameters used to calculate the retention indicator. Specifically, the choice of the time of ice shell formation as the freezing time needs justification.

   Unless the points raised in the foregoing can be shown to be unimportant, the Conclusion section should include less definite statements about complete retention in clouds.

   Minor points:

   Unless already well embedded in the literature, the terms "riming-retention" and "freezing-retention" should be reconsidered. The latter could apply to both riming (small droplet) and raindrops. The 'droplet' vs. 'drop' distinction is generally accepted in the literature and although imperfect as a definition it may be better to use the terms 'retention in freezing droplets' and 'retention in freezing drops'. Unfortunately, while it would be useful, it is impractical to also include in the terms some indication of what is being retained. Maybe acronyms have to be relied on.

line 18-20:  suggest using " ...aerosols from the boundary layer .." and " ... troposphere, and that can alter ...."

line 58: suggest 'visualize' instead of 'conceptualize'

line 62-62: suggest to replace 'infer a more systematic understanding' with a simpler 'improve understanding'

line 67: omit  'which was'

Eqn (3) might add the explicit result combining (1) and (2). Also would be informative to get some idea of the magnitude of D for the experiments for different temperatures.

line 143 and others: it would better to avoid the phrase 'freezing profiles' as there are too many different contexts for freezing already. Perhaps 'temperature graph' or just 'temperature] could be used. Even less useful is 'INP freezing profile'.

References:

Kleinheins, J., A. Kiselev, A. Keinert, M. Kind, and T. Leisner, 2021: Thermal Imaging of Freezing Drizzle Droplets: Pressure Release Events as a Source of Secondary Ice Particles. J. Atmos. Sci., 78, 1703-1713. https://journals.ametsoc.org/view/journals/atsc/78/5/JAS-D-20-0323.1.xml.

Korolev, A., and T. Leisner, 2020: Review of experimental studies on secondary ice production. Atmos. Chem. Phys. Discuss., 2020, 1-42. https://www.atmos-chem-phys-discuss.net/acp-2020-537/.

Lauber, A., A. Kiselev, T. Pander, P. Handmann, and T. Leisner, 2018: Secondary ice formation during freezing of levitated droplets. Journal of the Atmospheric Sciences https://doi.org/10.1175/JAS-D-18-0052.1.

List, R., 2013: New Hailstone Physics. Part I: Heat and Mass Transfer (HMT) and Growth. J. Atmos. Sci., 71, 1508-1520. http://dx.doi.org/10.1175/JAS-D-12-0164.1.

Field, P. R., and Coauthors, 2017: Secondary Ice Production: Current State of the Science and Recommendations for the Future. Meteorological Monographs, 58, 7.1-7.20. https://journals.ametsoc.org/view/journals/amsm/58/1/amsmonographs-d-16-0014.1.xml.

---

## Author Response (AR1)

We are grateful to both the reviewers for taking the time in going through our manuscript and provide insightful feedback and comments regarding our study. We have carefully addressed the reviewers' comments and suggestions in the responses provided below. **Red colored** text indicating **reviewers' comments**, and **black font** indicating **our responses** to reviewers' comments. Rewritten and newly added texts in the manuscript are provided below in *italics* for convenience. A revised version of the manuscript will be uploaded for the handling editor's consideration.

**The following are our responses to comments from the first reviewer- RC1:**

**RC1:** This manuscript presented the retention coefficients of trace gases in the freezing rain size droplets. This study used acoustic levitator to freeze droplets initiated by silver iodide and then measured the remained substances in the frozen droplets. The pH and temperature dependences are also investigated. New parameterization of the retention indicator are proposed. The topic of this study fits the scope of this journal. This manuscript is well written. There are several issues need to be addressed before publication.

 **Major comments:**

1. **RC1:** L35-58, do we expect significant difference in retention of volatile gases regarding the physical mechanism between cloud and rain droplets? A better rationalization focusing on rain droplet size is needed.

    **Response:**

    Yes, we do expect a difference between the retention of μm sized cloud droplets and mm sized rain drops. Firstly, the freezing initiation is different for cloud droplets where they freeze upon contact with a frozen substrate. The geometry would also change for droplets upon contact with frozen substrates. Due to ventilation in the experiments involving cloud droplets, spreading factor upon contact has to be accounted for, influencing the heat transfer into ice. The surface temperature of the freezing droplets on ice surface would also be warmer than the ambient temperature.

    Incase of our current experimental setup, freezing is initiated via ice nucleating particles, without any contact from an additional hydrometeor. Geometry and shape are not affected during the freezing.

    Secondly, the surface to volume ratio is three orders of magnitude higher for cloud droplets, which would affect the mass transfer timescale. Smaller droplets would have faster solute mass expulsion time.

    Text has been added in the manuscript in L54-59 as:

*"A significant difference from a physical perspective in terms of retention of trace gases for cloud droplets and rain drops would be the initiation and pathway of freezing. For riming experiments involving cloud droplets freezing is initiated upon contact with a frozen substrate, whereas, for rain drops investigated in this present study, immersion freezing was implemented. The geometry of the droplets upon contact also changes leading to spreading of the droplets under ventilated conditions in the riming-retention experiments. This change in geometry influences the heat transfer into the ice as it freezes. Moreover, the surface to volume ratio for cloud droplets is about 3 orders of magnitude higher as compared to rain drops. This higher surface to volume ratio would facilitate faster mass expulsion time for cloud droplets."*

2. **RC1:** L95, this study only investigated at concentration of 20 mg/L. What are the typical concentrations of investigated substances in the real rain drops? Are the retention coefficients also depending on the initial concentrations?

   **Response:**

   Typical concentration of trace gases lies in the range of ppb to tens of ppm. 1 ppm corresponds to 1 mg/L. Yes, in our case retention coefficients do depend on the initial concentrations as we have non-equilibrium conditions in our experiments. However, in nature, one can assume the trace gases to be in equilibrium in the atmosphere.

   Firstly, using higher than typical concentrations is due to the detection limit of the Ion chromatography (IC) column. One sample of 10 collected frozen drops corresponds to a volume of 40 µL. The minimum injection volume of the IC is about 250 µL. Filtration of samples is another necessary step to safeguard the IC column. During filtration about 50 µL of the total injection volume is also used up. Hence, the initial sample volume of 40 µL is diluted by about 9 times to obtain a clear signal in the chromatograph. The need for dilution during our quantitative measurement phase led us to opt for a higher initial concentration as compared to typical atmospheric conditions.

   High initial concentration of 20 mg/L would also imply that the internal partial pressure of any dissolved substances would be high enough to overcome the internal resistances inside the liquid drop. Higher concentrations used in our experiments would serve as the upper limit for minimum possible retention of the dissolved substances. But in this case retention is close to 1. This means lower concentrations would have also retentions of 1.

   Text has been added in the manuscript in L102-107 (previously L95) as:

*"Typical concentration of dissolved gases in the atmosphere lies in the range of ppb to tens of ppm. This higher concentration of 20 mg/L helped us maintain proper detection levels during our quantitative analysis. A high initial concentration of 20 mg/L would also imply that the internal partial pressure of any dissolved substances would be high enough to overcome the internal resistances inside the liquid drop. Higher concentrations used in our experiments would serve as the upper limit for minimum possible retention of the dissolved substances."*

3. **RC1:** Section 3.2, for higher pH, NaOH is used to control the pH of the solution. There is no $H^+$ for the investigated acids to be partitioned into gas phase, so of cause the retention coefficient would be close to 1 even without freezing. Please comment on this.

   **Response:**

   We agree. Increasing pH would modify the availability of H+ ions. At pH 7 there would be 3 orders of magnitude lower H+ as compared to pH 4. But still there would be H+ ions remaining in liquid phase. Lowering of H+ ions would affect effective Henry's Law coefficient, H*. For example, in the case of formic acid, H* increases by 2 orders of magnitude at ph 7 compared to ph 4. This is true also for smaller cloud droplets. With higher H* values, the retention coefficient is expected to shift towards right hand side of the sigmoid behavior seen in Fig. 4 in the manuscript. However, for large drops, gaseous and aqueous phase diffusivity were found to be the controlling factor for higher retention values (see Table 3, in the manuscript).

**Other comments:**

1. **RC1:** Title, Retention of what? It is suggested to include the subjects in the title. It would be more specific.

   **Response:**

   Title has been changed for better clarity. New title: "*Retention During Freezing of Raindrops, Part I: Investigation of Single and Binary Mixtures of Organic and Inorganic Trace Gases*"

2. **RC1:** L8, which physical properties?

   **Response:**

   The physical properties referred here are the drop size and ice-shell formation during freezing, which has been explained further in L10. Text has been added in L8 as:

*"Our results show the dominance of physical aspects such as drop size and ice shell formation over their chemical counterparts on overall retention for the investigated large drops."*

3. **RC1:** L50, which studies mentioned? Please be specific.

   **Response:**

   Here we refer to all the above-mentioned studies, as all the studies were focused on µm sized cloud droplets. Text has been changed in L51 (previously L50) as:

   *"All the above mentioned experimental studies concerning riming-retention were mostly related with cloud sized droplets (i.e. diameters in the µm size range), for which the chemical properties were determined to be the dominant factors."*

4. **RC1:** Section 2.1, what are the temperature uncertainties?

   **Response:**

   The temperature mentioned here corresponds to the temperature of the cold room. The uncertainty of the cold room temperature is about ± 1°C. However, this uncertainty is not important, provided that we are recording the drop surface temperature in our measurements via an infrared thermometer. All the analysis has been made with regard to drop freezing temperatures, recorded via the infrared thermometer. The uncertainties concerning the drop freezing temperatures have been mentioned explicitly throughout the manuscript.

   Accordingly, the text has been changed in L91 for better clarity as:

   *"We first characterized the INP at three different concentrations (0.2, 0.01, and 0.0003 g/L) at three different cold room temperatures (-15, -20, and -28 °C)."*

5. **RC1:** L352, temperature at $f_{ice}$ of 50% is the median freezing temperature, not mean/average.

   **Response:**

   Thank you for pointing it out. Text has been changed in L399 (previously L 352) as:

   *"The temperature at which $f_{ice}$ was 50% was taken as the 'median drop freezing temperature' for each set of concentration and cold room temperature pair."*

6. **RC1:** Missing page information for some references.

   **Response:**

   Thank you for pointing it out. Page information has been added for the missing references.

**The following are our responses to comments from Gabor Vali- RC2:**

**RC2:** Referee comment by Gabor Vali on "Retention During Freezing of Raindrops, Part I: Investigation of Single and Binary Mixtures" by Gautam et al.

This manuscript extends to raindrops the work previously done by the authors and their colleagues on retention coefficients for cloud droplets. Acoustic levitation in a cold room and chemical analyses before and after freezing constitute the essence of the experiments. The levitation system and the use of infrared thermometry avoid the need for contact with any support. This is a near ideal arrangement. The paper presents a good description of the experiments and sound analyses of the retention coefficients. The paper is well constructed and well written (with a few odd phrasing). The topic is quite appropriate for ACP.

This reviewer has not been involved for decades with the topic of retention of foreign material from ice growth and will only address in these comments the physical aspects of the experiments, how to interpret the results, and to what extent the results apply to processes in atmospheric clouds.

**RC2:** Two features of the experimental approach are the focus of these comments: the large difference in temperature between the drop and the surrounding air, and the near-absence of ventilation.

The 2-mm diameter raindrops used in the experiments are up to 100 times larger than the cloud droplets used in the previous experiments and thus have about 100 times larger volume to surface ratios. That would lead one to anticipate significantly slower freezing and, consequently, larger rejection of foreign substances as ice forms. The results here presented show the opposite. For two of the substance involved in both experiments (formic and acetic acids), values near 0.7 were obtained in the riming experiments and near 1.0 for the raindrops. The authors' chief argument for this is that the larger drops in free air had an ice shell form on their outside trapping most of foreign substances.

The formation of the ice shell after nucleation is well documented in the paper. It is also what one would expect for an isolated drop with the air temperature considerably lower than the drop temperature even before nucleation and pronouncedly so during the freezing of the drop when the surface temperature rises to near 0ºC (Fig. A1). In contrast, in the atmosphere, the temperature of the drop would be close to the air temperature before freezing. It would also have asymmetric heat transfer when nucleation and initial ice formation leads to latent heat release within the drop. The resulting surface temperature and the formation of ice within the drop will be influenced by the asymmetry and by the rate of heat transfer to the environment. Theoretical analyses of the problem have been made with respect to hailstone formation and growth (e.g. List, 2014). These analyses also consider evaporation from the drop surface and collection of cloud droplets, but do not treat explicitly how ice forms inside the drop. For the current

discussion, more relevant are the many experiments, and drops caught in clouds, that demonstrate that frozen drops often have protuberances and other deformation on their surfaces.

**Response**:

We agree with the reviewer's comment, that the heat transfer and the temperature difference between the freezing drop and the environment may influence the freezing and retention. In addition, also an asymmetric flow field affects the freezing and its rate. However, in our understanding freezing takes place in two freezing stages: the adiabatic freezing stage and the diabatic freezing stage (Pruppacher and Klett, 1997). The adiabatic freezing accounts for rapid crystal growth, where only a small fraction of liquid freezes and majority of latent heat released during cooling contributes to warming the supercooling drop to 0°C (Stuart and Jacobson 2003, Szakall et al., 2021). We assume that ice shell formation occurs right after or in between the adiabatic freezing stage, which is time independent and where no exchange with the environment occurs. Thus, we expect that this freezing stage should not be affected by the temperature difference between the drop and the ambient air.

The adiabatic freezing time recorded by the infrared thermometer with a resolution of 0.5 s in our experiments, was about 1 s. The freezing of supercooled drops can be divided into two stages, diabatic and adiabatic (Pruppacher and Klett, 1997). The adiabatic freezing accounts for rapid crystal growth, where only a small fraction of liquid freezes and majority of latent heat released during cooling contributes to warming the supercooling drop to 0°C (Stuart and Jacobson 2003, Szakall et al., 2021). Ice shell formation takes place soon after adiabatic freezing stage. And dDiabatic freezing is where heat exchange takes place between the drop and ambient air (also shown in Fig A1, Appendix A1 in the manuscript). Definitely the temperature difference and ventilation do influence the diabatic freezing rate and deformation, as well as potential for crack formation. A typical total freezing time of our drops is 50 s. When we compare this with the results from numerical calculations of Stuart and Jacobson (2006), using a ventilated 2 mm drop falling at terminal velocity it is approximately 60 s at -5 °C, 30 s at -10 °C, and 15 s at -20 °C, ambient temperature (700 hPa). This shows that for temperatures similar to those in our experiments the total freezing time for a fully ventilated drop is lower by a factor of about 4. Ice shell formation should have occurred even faster as in our experiments. A faster ice shell formation means a higher retention of the species. Hence, our results can be assumed as a lower limit for retention. However, for our experimental setup with low ventilation coefficient of approximately 5 the retention values were already close to 1, even for very volatile species. All species are impeded by the aqueous phase mass transport, meaning that when we increase heat transport also the mass transport should be enhanced. But if we have the aqueous diffusion as a limiting factor, mass transport of the species to the environment should be limited. This higher ventilation means higher retention. We do believe that naturally falling rain drops, freeze faster than in our case which would lead to retentions of 1.

We went through the theoretical heat transfer analysis in List (2014), however it is not completely applicable in our case of a motionless water droplet in air. There is a near absence of evaporation and heat transfer due to accretion of cloud droplets, for our experimental set up because of the short time (approx. 10 to 20 sec) the drop levitates in the acoustic field before freezing. Equation(3) in List (2014) is most relatable to our case, where they describe heat transfer by conduction and convection. But factors accounting for turbulence and surface roughness described in Eq(3), List (2014), would not be very realistic in our case. Moreover, due to the presence of ultrasonic field while levitating the drop, precise measurements for heat and mass transfer cannot be evaluated. A simplified theoretical estimation was reported for the heat and mass transfer was described in Appendix B1, Szakall et al., (2021), where they integrated the heat flux density over the entire drop surface. They considered the diffusional heat transfer as a function of drop radius, while latent heat from condensation and evaporation was not included for the case of isolated levitating drop. In reality, a static heat from the ultrasonic field also exists which was not considered in their heat transfer derivation as the contribution from ultrasonic heating effect was negligible at subzero temperatures. Overall, the changes in drop surface temperature measured from the infrared thermometer accounted for influences from the ultrasonic field collectively. Calculating the thermal conductivity in air from Eq(13.18a) in Pruppacher and Klett (2010), the heat transfer rate for a 2 mm drop at a temperature of -15°C was found to be $2.01*10^{-3}$ Joules/s.

Text has been added in L334 as:

*"The temperature difference between the freezing drop and the environment may influence the freezing and retention. During the initiation of freezing the drop temperature rises to 0∘C (FigA1), where fraction of liquid freezes and majority of latent heat released during cooling contributes to warming the supercooling drop to 0°C (Szakáll et al., 2021). We assume that ice shell forms very rapidly at this stage, which can be perceived as adiabatic freezing (see A1) with no exchange of heat to the environment. Thus, we expect that this freezing stage should not be affected by the temperature difference between the drop and the ambient air"*

**RC2:** Cracks in the ice shell may lead to the expulsion of liquid to the surface and perhaps to the air. The theory of ice multiplication in clouds by splintering is based on those observations (Field et al, 2017; Lauber et al., 2018). The potential for cracks in the ice shell may also have to considered for the experiments described in the paper. Internal pressure rises as the drop freezes and is likely to produce cracks in the ice shell (e.g. Korelev and Leisner, 2020; Kleinheins et al., 2021; references herein). Because of the low air temperature in the experiments, any excluded water is likely to freeze onto the surface quite rapidly. This would slow internal freezing. The cited papers describe work with water without added substances. Dissolved gases or ions may modify the freezing behavior.

**Response:**

Thank you for the valuable comment. We calculated the internal pressure inside the liquid drop for our experimental setup, using Eq(3) in Kleinheins et al., (2021). The drop sizes in our study are larger by an order of magnitude compared to the ones reported in Kleinheins et al., (2021). For our temperature graph shown in Fig A1, the highest temperature deviation can be seen at about 55 seconds. Using Eq(3) from Kleinheins et al., (2021), the internal pressure was found to be by 80.76 bar for a freezing point depression of 0.6 K. This internal pressure is quite less for splitting events to occur, which took place well above 100 bars as reported in Kleinheins et al., 2021(their Fig 6, Experiment type A: Stagnant air). This eliminates the potential for cracks in the ice shell during the freezing and subsequent source for secondary ice production with our existing setup, as mentioned in Field et al, (2017) and Lauber et al., (2018). Furthermore, fluctuations in our temperature graph could also be due to deformations from the spherical shape of the levitated drops as is undergoes diabatic freezing. The semi frozen drop protrudes in the vertical direction and aligns itself according to the ultrasonic field, thus, influencing the emissivity measured by the infrared thermometer. The internal pressure built up during freezing of the levitated drop in our experiment might not be enough for eclosed liquid and or solute to be expelled outside. We did observe rapid melting and refreezing of water (in the vertical direction, on the surface) as the semi-frozen drop tried to expand and align itself along the acoustic field. This can potentially lead to expelling molecules from the drop, i.e., working against retention. This rapid melting and refreezing events took about 2.6ms, perhaps not long enough for dissolved solute to be expelling out from the drop. Further investigation into the crack propagation and internal pressure build up for our experimental setup via high speed imaging is ongoing.

 Text has been added in L328 as:

"*Observations of naturally frozen drops and laboratory experiments (Lauber et al., 2018) have shown frozen rain drops having deformations and protuberances. Rise in internal pressure during freezing could have potential for cracks or splitting of ice shell during freezing, leading to expulsion of solute mass and perhaps source for secondary ice production (Field et al., 2017; Korolev and Leisner, 2020). Kleinheins et al. (2021) reported cracking of ice shell for internal pressure above 100 bar for 300 µm sized drops. The internal pressure built up during freezing in our experiments was found to be about 81 bar. Within our experimental conditions, these occurrences were however not seen.*"

**RC2:** Most of the foregoing work was done with droplets of hundreds of micrometer in diameter, not far but still below the size of 2 mm involved in the current experiments. That discrepancy and the complex nature of the phenomenon make any extrapolation difficult and it is even more speculative how all of the above influence retention of foreign substances. In that light, it is a welcome development to have the results presented in

this paper. However, it is clear that more work is needed and that the authors of this paper should express their views on the matter in the manuscript.

**Response:**

We agree. The presence of foreign substances could affect the freezing as well. And as mentioned above, further work is undergoing to understand the complexity and the factors affecting the freezing process for an isolated and freely levitating drop.

Presence of ventilation in our experiments could increase the internal circulation inside the levitated drops, a occurrence similar to the case of an ultrasonic bath. Increased circulation could lead to an increase in internal pressure as well. We are currently modifying our setup to investigate the effect of ventilation with high speed camera setup and observe any potential cracks or liquid expulsion events. We are also undertaking a detailed investigation of the ice shell propagation and subsequent ice formation inside the supercooled drop, after the ice shell is formed. However, the results are preliminary and an in-depth investigation is required and ongoing.

For the case of dissolved substances, yes, they could lower the growth of ice crystals or freezing rate for solute concentrations (Pruppacher 1967). The solute concentration in our study was about 20 ppm, which corresponds to about $4.3*10^{-4}$ moles/L for formic acid (substance with the least molar mass among the investigated chemical species). At such concentrations, the influence of dissolved substances on the freezing could be considered rather negligible.

Text has been added in L107 as:

*"Additionally, high solute concentration could decrease the freezing rate (Pruppacher, 1967). For 20 mg/L, the effect of dissolved substances influencing the freezing process could be considered negligible as compared to pure water. In terms of molar concentration, 20 mg/L corresponds to $4.34 \times 10^{-4}$ moles/L for formic acid - which has the least molar mass among the investigated species - is 2 orders of magnitude lower to significantly affect the freezing process (Pruppacher, 1967)."*

**RC2:** Another dimension of the problem is how the high retention found in this work might be envisaged on the molecular scale. Some discussion of the results of molecular simulations of crystal growth may help readers' understanding of the results.

**Response:**

Thank you for pointing it out. In our experiments we have fast freezing rates which implies the molecules do not have time to diffuse away from the forming ice front. This means the molecules are easily captured by the ice and form defects in the ice crystal lattice. Stuart and Jacobson (2006) reported the formation of liquid pockets that can trap solutes during freezing from their numerical simulation, informed from previous studies of dendritic

crystal growth in solution. Formation of liquid pockets was also seen during the diabatic freezing stage in our experiments after the ice shell has been formed. The ice shell formation impedes further retention because diffusion in ice is orders of magnitude lower compared to the liquid. Moreover, the liquid-to-ice partitioning coefficient—indicating the extent to which solutes are incorporated into the ice—is relatively high for malonic and acetic acids, particularly at high freezing rates, often exceeding 0.8 (Hey, M., 2022). As a result, only a small fraction of the solute is expelled. This high degree of solute incorporation into the ice is the primary factor contributing to the observed retention.

We did try to simulate the numerical model from Stuart and Jacobson (2006) for our experiments to further investigate on a molecular level. We attempted to build upon the existing model to match our setup, by adjusting the parameters such as ventilation, Reynolds number, aqueous and gaseous diffusivities, Henry's law coefficients for each chemical substance, but it did not turn out to be very representative of our experimental results. Hence, we proceeded with a simpler retention indicator timescale investigation approach for our experiments, as described in Stuart and Jacobson (2003) and Jost et al., (2017).

Text has been added in L321 as:

*"The fast freezing rates observed in our study implies that the molecules do not have much time to diffuse away from the forming ice front. This means the molecules are easily captured by the ice and form defects in the ice crystal lattice. Stuart and Jacobson (2006) reported the formation of liquid pockets that can trap solutes during freezing from their numerical simulations, informed from previous studies of dendritic crystal growth in solutions. These liquid pockets were also seen in our experiments. The ice shell formation impedes further retention because diffusion in ice is orders of magnitude lower compared to the liquid. As a result, only a small fraction of the solute is expelled. This high degree of solute incorporation into the ice is the primary factor contributing to the observed high retention in our study."*

**RC2:** Section 3.5 deals with the retention indicator defined by the relative timescales of mass expulsion and that of freezing. It is unclear if this measure is intended to describe and idealized freezing front or is applied to specific geometries, spheres in this case. Perhaps the authors can illuminate this by justifying their choose of the parameters used to calculate the retention indicator. Specifically, the choice of the time of ice shell formation as the freezing time needs justification.

**Response:**

Retention indicator is the ratio between the expulsion timescale and the freezing timescale which can be divided into the adiabatic and diabatic phase. The expulsion timescale is well established in atmospheric chemistry applications (Seinfeld and Pandis, 2006, Schwartz, 1986) to describe the uptake of gases by droplets. We have

adjusted it for ventilated conditions. It approximates the mass transfer similar to the resistances in an electrical circuit. The freezing time assumes that the ice front propagates radially through the spherical drop or in case of riming (Jost et al, 2017) in an approximated cylindrical geometry. Also, previous studies have utilized the retention indicator, showing it to be an effective parameter for describing retention as shown in Stuart and Jacobson (2003) and Jost et al., (2017).

The geometry of the freezing particle is taken into consideration in our case. The freezing timescale analysis in Jost et al., (2017) considered a cylindrical shape factor for cloud droplets for their experiments. This shape factor has been adjusted for spherical case for our timescale calculations. The choice of gaseous, aqueous diffusivity, mass transfer parameters were taken from Stuart and Jacobson (2003) and Jost et al., (2017), which were reported to be the most probable controlling factors for retention of solutes during freezing.

In our present study we found that once the ice shell is formed during the adiabatic freezing stage, it does not create cracks/splits in the diabatic freezing stage. Thus, the ice shell prevents any further expulsion of solute and liquid mass until the drop is completely frozen. Stuart and Jacobson (2003, 2006) also state that if an ice shell is formed immediately after adiabatic freezing stage, it limits the solute transfer and impedes further expulsion into the gas phase. Hence, the ice shell formation time during the adiabatic freezing stage has been considered as the freezing time for our retention indicator analysis.

Text has been added in L285 as:

*"The ice shell formation takes place very rapidly during the adiabatic freezing stage of the drop, where drop surface temperature rises to 0∘C (see A1). After the formation of the ice shell the dissolved solute remains inside and retained during freezing. Hence, this ice shell formation time was considered as the freezing time for the retention indicator calculation."*

**RC2:** Unless the points raised in the foregoing can be shown to be unimportant, the Conclusion section should include less definite statements about complete retention in clouds.

**Response:**

Changes and text have been added as mentioned in the responses above.

Text has been changed and added to the conclusion section, in L354 as:

*"This indicates that during the freezing of mm sized raindrops all dissolved trace gases may be removed by precipitation in deep convective clouds or transported within the ice phase into the UT where it can be released upon sublimation. Concurrently, factors such*

*as ventilation, temperature differences, crack formation during freezing and concentration of dissolved solute needs to be dealt meticulously. Our results, combined with results from riming-retention studies facilitates the extrapolation of retention of the investigated trace gases from μm to mm sized drops in computational studies."*

And in L 369 as:

*"Furthermore, in-depth investigation of the effect of ventilation and examination of internal pressure build up during freezing for rain drops also provides an interesting aspect to investigate in future studies."*

**Minor points:**

**RC2:** Unless already well embedded in the literature, the terms "riming-retention" and "freezing-retention" should be reconsidered. The latter could apply to both riming (small droplet) and raindrops. The 'droplet' vs. 'drop' distinction is generally accepted in the literature and although imperfect as a definition it may be better to use the terms 'retention in freezing droplets' and 'retention in freezing drops'. Unfortunately, while it would be useful, it is impractical to also include in the terms some indication of what is being retained. Maybe acronyms have to be relied on.

**Response:**

The term riming retention was considered to clearly distinguish between the freezing processes involving cloud droplets and raindrops, and in order to avoid any confusion between previous studies involving μm sized cloud droplets and current study involving larger mm sized drops.

After discussion with our coauthors, we have decided to hold on to our current notations. Droplet vs drop distinction is quite well known in the cloud physics community, but perhaps not so in atmospheric chemistry community.

**RC2:** line 18-20: suggest using " ...aerosols from the boundary layer .." and " ... troposphere, and that can alter ...."

**Response:** Changes have been made in L20

**RC2:** line 58: suggest 'visualize' instead of 'conceptualize'

**Response:** Changes have been made in L65 (previously L58).

**RC2:** line 62-62: suggest to replace 'infer a more systematic understanding' with a simpler 'improve understanding'

**Response:** Changes have been made in L68 (previously L62).

**RC2:** line 67: omit 'which was'

**Response:** Changes have been made in L74 (previously L67).

**RC2:** Eqn (3) might add the explicit result combining (1) and (2). Also would be informative to get some idea of the magnitude of D for the experiments for different temperatures.

**Response:**

Magnitude of desorption D, in our study has already been uploaded in our data repository. With the thought of avoiding any potential point for confusion to the readers, the magnitude of D was not included in the main manuscript.

Text has been added in L153 to highlight this point, as:

*"The experimental data for retention coefficients of the investigated species and their desorption can be accessed at Gautam and coauthors (2024)."*

**RC2:** line 143 and others: it would better to avoid the phrase 'freezing profiles' as there are too many different contexts for freezing already. Perhaps 'temperature graph' or just 'temperature] could be used. Even less useful is 'INP freezing profile'.

**Response:** Thank you for pointing it out. The changes have been made in the manuscript accordingly.

**References (RC2):**

Kleinheins, J., A. Kiselev, A. Keinert, M. Kind, and T. Leisner, 2021: Thermal Imaging of Freezing Drizzle Droplets: Pressure Release Events as a Source of Secondary Ice Particles. J. Atmos. Sci., 78, 1703-1713. https://journals.ametsoc.org/view/journals/atsc/78/5/JAS-D-20-0323.1.xml.

Korolev, A., and T. Leisner, 2020: Review of experimental studies on secondary ice production. Atmos. Chem. Phys. Discuss., 2020, 1-42. https://www.atmos-chem-phys-discuss.net/acp-2020-537/.

Lauber, A., A. Kiselev, T. Pander, P. Handmann, and T. Leisner, 2018: Secondary ice formation during freezing of levitated droplets. Journal of the Atmospheric Sciences https://doi.org/10.1175/JAS-D-18-0052.1.

List, R., 2013: New Hailstone Physics. Part I: Heat and Mass Transfer (HMT) and Growth. J. Atmos. Sci., 71, 1508-1520. http://dx.doi.org/10.1175/JAS-D-12-0164.1.

Field, P. R., and Coauthors, 2017: Secondary Ice Production: Current State of the Science and Recommendations for the Future. Meteorological Monographs, 58, 7.1-7.20. https://journals.ametsoc.org/view/journals/amsm/58/1/amsmonographs-d-16-0014.1.xml.

**References (AC1):**

Hey, M.: Experimentelles Bestimmen der Verteilungskoezienten wässriger Malonsäurelösungen: *Experimental determination of partition coefficients of aqueous malonic acid solution, Bachelor thesis*, University of Mainz, Germany, 2022.

Jost, A., Szakáll, M., Diehl, K., Mitra, S. K., and Borrmann, S.: Chemistry of riming: the retention of organic and inorganic atmospheric trace constituents, Atmospheric Chemistry and Physics, 17, 9717–9732, 2017.

Pruppacher HR. Some relations between the structure of the ice-solution interface and the free growth rate of ice crystals in supercooled aqueous solutions. Journal of Colloid and Interface Science. 25(2):285-94, 1967.

Pruppacher, H. R. and Klett, J. D.: Microstructure of atmospheric clouds and precipitation, Microphysics of clouds and precipitation, Ch. 11, 2010.

Schwartz, Stephen E. "Mass-transport considerations pertinent to aqueous phase reactions of gases in liquid-water clouds." Chemistry of multiphase atmospheric systems. Berlin, Heidelberg: Springer Berlin Heidelberg, 415-471, 1986.

Seinfeld, John H., and Spyros N. Pandis. Atmospheric chemistry and physics: from air pollution to climate change. John Wiley & Sons, 2016.

Stuart, A. L. and Jacobson, M.: A timescale investigation of volatile chemical retention during hydrometeor freezing: Nonrime freezing and dry growth riming without spreading, Journal of Geophysical Research: Atmospheres, 108, 4178-4194, 2003.

Stuart, A. L. and Jacobson, M. Z.: A numerical model of the partitioning of trace chemical solutes during drop freezing, J. Atmos. Chem., 53, 13–42, 2006.

Szakáll, M., Debertshäuser, M., Lackner, C. P., Mayer, A., Eppers, O., Diehl, K., Theis, A., Mitra, S. K., and Borrmann, S.: Comparative study on immersion freezing utilizing single-droplet levitation methods, Atmospheric Chemistry and Physics, 21, 3289–3316, 2021.475

---

## Editor Decision (ED1)

Main editor comments: (Line numbers refer to the manuscript version with track-change).

1) Title: The referee asked 'Retention of what'?
I think the changed title still doesn't answer the question. Wouldn't it be clearer to say

*Retention of Organic and Inorganic Trace Gases during Freezing of Rain Drops: Part 1 Investigation of Single and Binary Mixtures.*
Or even
*Retention of nitric, formic and acetic acids and nitrophenol….*

I am aware that you have a 'Part II' paper in review as well. However, you may consider changing its title accordingly.

2) I am confused about your discussion on the effective Henry's constants, e.g. in sections 3.1 and 3.4 but also in the introduction where you refer to previous studies. You say that in previous studies, compounds showed a dependence on pH due to increasing solubility ($K_H^*$) with increasing pH.
I argue that some of the observed pH dependencies are not a function of solubility or KH*. For example, within the pH range of your experiments (3 < pH < 6), KH* for nitrophenol does not change (see figure below – note that, for simplicity, I used values at 25C and expressed KHeff in M/atm; however, the trends are likely similar at lower T, and are independent of the unit). Therefore, if retention were a function of solubility, no change in R should be expected. The same applies for acetic acid at pH < ~5.

[Figure]

If I understand correctly, you used your measured R values and KH* values to derive the empirical coefficients a and b in equation 4. Which R values did you use as input to the equation? In your Fig 2, you show that there is a statistically significant difference for R(nitrophenol) as a function of pH – while KH* is identical.
It would be useful if you commented on this and maybe even include a figure as above (as a supplement) to add to the discussion.

3) There are several studies that revealed that the gas-aqueous partitioning of nitrophenols in cloud droplets may not adhere to their Henry's law constants, e.g.
Lüttke et al., Phenols and Nitrated Phenols in Clouds at Mount Brocken, Intern. J. Environ. Anal. Chem.. Vol. 74(1-4). pp. 69-89
Lüttke et al., Phase partitioning of phenol and nitrophenols in clouds, Atmos. Environm., 1997, 2649-55.
I wonder if the behavior of nitrophenol in your study could be partially explained by this.

Minor/Technical comments
Please carefully proofread the paper. In particular pay attention to the correct use of articles.
a) I list a few places below where 'the' or 'a' is missing, e.g.
l. 20: from the boundary layer
l. 36: in the context
l. 331: could have the potential
l. 333: cracking of the ice shell
l. 337: where the fraction of liquid freezes and the majority …

b) Also please pay attention to the consistency of singular/plural forms of subject and verb, e.g.
l. 61: Freezing of raindrops is…
Table 2: …temperature was
l. 323: …rates … imply

Abstract:
l. 1/ 2: You may want to consider improving the first sentence (in particular since it is the first sentence),
*The interaction with freezing processes and vertical transport of trace gases into the upper atmosphere during deep convection is critical to understanding the distribution of aerosol precursors and their climate effects.*

1) 'Interactions with..' does not seem right here
2) Processes do not really 'interact' – they may be coupled or influence/affect each other.

If I understand correctly, you want to say
"Freezing processes affect the vertical transport of trace gases into the upper troposphere…

l. 10: *"Thus, for rain sized drops almost everything is fully retained during the freezing process, even for species with low effective Henry's law constants."*
This sentence sounds quite colloquial. Given that you define 'retention coefficient', it may be clearer or more precise to say that the retention coefficients for all single compounds and mixtures were near 1 (or give a range)
Can you specify 'low effective Henry's law constant'?

l. 27: 'evident' seems redundant here

l. 30: Here you use 'drop' in the context of clouds – given the referee comment and your response, shouldn't it be 'droplet'?

l. 32, 37 (and maybe other places in the manuscript): For better readability, please move the references to the end of the sentence.

l. 43: 'Additionally' implies that H* is neither a chemical nor physical property as they were already mentioned in the previous sentence. Thus, 'additionally' seems redundant here.

l. 55: 'A significant difference from a physical perspective in terms of retention of trace gases for cloud droplets and rain drops would be the initiation and pathway of freezing'
- Why do you use subjunctive ('would')? If it is a well-known fact, 'is' is appropriate. (Please check the full manuscript for use of 'would' and decide whether the use of indicative ('is') is appropriate. )

- 'initiation and pathway of freezing' - is usually referred to as 'freezing mechanism'.

l. 57: 'was implemented' sounds odd. You may implement something in a model but this is certainly not meant here… isn't something like 'was the main mechanism' or 'took place' more appropriate?

l. 67: It may be useful to add the H* values here already, together with the pH value.

l. 69: what do you mean by '…values for riming with cloud droplet sizes' – is it simply 'in riming cloud droplets'?

l. 104: - ppm and ppb are mixing rations, not concentrations.
- Please specify that you mean 'ppb to ppm (on mass basis)' to make clear that you mean 1 g in 10^9 g or 10^6 g, i.e. 1 ug/L or 1 mg /L (assuming that water density = 1 g/cm3)
This avoids confusion since gas phase mixing ratios of trace gases are commonly given in ppb whereas e.g 50 ppb ozone means '50 molecules out of 10^9 molecules'

l. 111: 'least' should be 'lowest'

l. 120: 'benzoic' misspelled

l. 141: You used D already for drop diameter (abstract). I suggest changing it there and simply spell out 'drop diameter: 2 mm).'

l. 164/166: Even though Referee #1 did not specifically comment on the text in these lines, 'average freezing temperature' should be also replaced here by 'median freezing temperature'.

l. 169: Please clarify this sentence: " The 50% frozen fraction at-23∘C was found to be −6.9±1.1∘C."

l. 171: 'sized' can be omitted here and also in the remainder of the manuscript for similar instances.

Table 2: Clarify in the caption whether the R values are averaged over all pH values or only apply to a specific pH.

l. 182: "Brand (2014) studied the retention of large drops (2.67 mm and 7.25 mm spherical equivalent diameter)" – please clarify what Brand investigated. It should be the retention of gases (organic acids? All the same compounds as you used in the present study?) in large drops.

l. 185: 'with which …was realized' can replaced 'representing'

l. 193" replace 'least' by 'lowest'

l. 209-215: What is the main message here?
First you say that *"Acetic acid (green marker) and formic acid (blue marker) did not show any apparent dependency on pH"*
Then you say that "*The retention coefficients for acetic acid were 0.81, 0.88, and 1.05 for pH values of 3.1, 4.2, and 7.0, respectively, while their corresponding standard deviations were 0.18, 0.12, and 0.2"* – doesn't this trend show a dependence? I understand your argument that the standard deviations are

larger than the differences between the mean values – however, yet, the figure shows a clear trend and an average R at pH = 7 that is about 25% higher than that at pH = 3.1.

*"From Fig. 2a, one can infer a slight dependency on pH for 2-nitrophenol, and almost none for acetic acid and formic acid."*

I see it the opposite way based on the figure, i.e. that there is barely any dependence of R on pH for nitrophenol (at least at pH < 5); however, there is a steady increase of R with pH for acetic acid.

I understand your argument that your conclusions are based on the results of 11 experiments. Why don't you show these values rather than just the averages ± standard deviation that (falsely?) imply a trend and therefore contradict your text?

l. 219 - 224: It is difficult to understand what you are saying here.

*"pH of the solutions were altered by adding HCl and NaOH, which could also interact with the investigated substances and dissociate them into their ionic form"*

Isn't this idea of pH adjustment that you change the proportions of dissociated vs undissociated forms? I suggest omitting this sentence as the second part is confusing (if not even wrong as adding NaOH does not lead to dissociation but association of H+ and carboxylates), and the first part was already mentioned in Section 2.

*" In this case the overall concentration of the investigated substances could be lowered."*

Which concentration is lowered under what conditions? When acids dissociate (i.e. at enhanced pH) the total aqueous phase concentration (acid + anion) actually increases.

Or are you saying that the solubility of the solutes is expected to decrease in the presence of additional solutes such as HCl and NaOH due to salting-out effects? Are there any references for this? In such a case, the Henry's law constants for pure water may not be applicable.

*"After addition, the lowest measured initial liquid phase concentration was 17.8 mg/L (11% decrease)."*

Is this an expected trend or is this random variation due to evaporation of acids?

Table 3: Please indicate that you use dimensionless Henry's law constants.

l. 325/6: '*from their numerical simulations*' seems at a wrong place in the sentence. Please clarify.

l. 343: *"Our results show higher retention coefficients close to 1 for mm sized raindrops for similar substances from previously studied retention coefficients"*

This sentence should be restructured for clarity, e.g.

Our results show higher retention coefficients (close to 1) for similar substances in mm sized raindrops as compared to previously determined retention coefficients in um sized cloud droplets.

Section 4: Please make sure that the conclusion section adheres to the author guidelines at https://www.atmospheric-chemistry-and-physics.net/policies/guidelines_for_authors.html

---

## Author Response (AR2)

We are grateful for the editor's decision as well as taking the time to check our revised manuscript thoroughly. We have carefully addressed the editor's valuable comments and suggestions in the responses provided below. **Red colored** text indicating **editor's comments and suggestions**, and **black font** indicating **our responses** to them. ***Rewritten and newly added texts*** in the manuscript are provided below in ***italics*** for convenience. Line numbers mentioned here correspond to the revised manuscript. A revised version of the manuscript will be uploaded for the handling editor's consideration. **"Main editor comments: no.2, 1st part** and **"Minor/Technical comments: l. 209-215",** are highlighted for ease of navigation.

**Main editor comments**: (Line numbers refer to the manuscript version with track-change).

**Editor: 1)** Title: The referee asked 'Retention of what'?

I think the changed title still doesn't answer the question. Wouldn't it be clearer to say

*Retention of Organic and Inorganic Trace Gases during Freezing of Rain Drops: Part 1 Investigation of Single and Binary Mixtures.*

Or even

*Retention of nitric, formic and acetic acids and nitrophenol….*

I am aware that you have a 'Part II' paper in review as well. However, you may consider changing its title accordingly.

**Response:**

Considering the suggestions above, the title has been changed for better clarity.

*"Retention During Freezing of Raindrops, Part I: Investigation of Single and Binary Mixtures of Nitric, Formic and Acetic Acids and 2-Nitrophenol"*

**Editor: 2)** I am confused about your discussion on the effective Henry's constants, e.g. in sections 3.1 and 3.4 but also in the introduction where you refer to previous studies. You say that in previous studies, compounds showed a dependence on pH due to increasing solubility (KH*) with increasing pH. I argue that some of the observed pH dependencies are not a function of solubility or KH*. For example, within the pH range of your experiments (3 < pH < 6), KH* for nitrophenol does not change (see figure below – note that, for simplicity, I used values at 25C and expressed KHeff in M/atm; however, the trends are likely similar at lower T, and are independent of the unit). Therefore, if retention were a function of solubility, no change in R should be expected. The same applies for acetic acid at pH < ~5.

[Figure]

*Figure A. Relationship between effective Henry's constant and pH. (We labelled this figured for the ease of reference)*

If I understand correctly, you used your measured R values and KH\* values to derive the empirical coefficients a and b in equation 4. Which R values did you use as input to the equation? In your Fig 2, you show that there is a statistically significant difference for R(nitrophenol) as a function of pH – while KH\*is identical. It would be useful if you commented on this and maybe even include a figure as above (as a supplement) to add to the discussion.

**Response:**

We are greatly thankful for your insightful comment and the plot (Fig. A). We have included a similar plot in the supplement as Figure S1.  We are splitting our responses into two parts: **1ˢᵗ part** - regarding the 'observed pH dependencies are not a function of solubility or KH\*' and **2ⁿᵈ part** - regarding equation 4, (a and b parameters).

We also found the comments regarding the **1ˢᵗ part** here, concerning KH\* and pH, and in **"Minor/Technical comments: l. 209-215",** concerning pH dependencies to be closely related to each other.  Hence, our responses for both these similar sections are collectively stated here, to avoid repetition.

**1ˢᵗ part:**

**(and "Minor/Technical comments: l. 209-215:")**

We agree to the statement "some of the observed pH dependencies are not a function of solubility or KH\*". From fig. A, one should ideally observe a pH dependence for formic and acetic acid and none for 2-nitrophenol for our measured pH range. Our ignorance in

performing a proper statistical analysis for pH dependencies, led to misinterpretation of our data.

We did a careful investigation of our data and performed linear regression test (95% confidence interval) using IBM SPSS Statistics-Version23 for the investigated single components and their dependence on pH values. p- value lower than 0.05 indicates dependence of retention coefficients on pH. The results for single component substances are as follows:

Acetic acid:

The linear regression tests reveal a significant statistical dependence of the retention of acetic acid on pH, with p = 0.047.  This result contradicts what we had previously assumed for dependence of acetic acid on pH (owing to large standard deviations). The linear regression test results are in agreement with "==Minor/Technical comments: l. 209-215:== 'I see it the opposite way based on the figure, i.e. that there is barely any dependence of R on pH for nitrophenol (at least at pH < 5); however, there is a steady increase of R with pH for acetic acid.'  Acetic acid is not completely retained at pH 4, so an increase in retention can be seen at higher pH which is also supported by Fig. A.

**Text has been added/changed in L206 as:**

*"Linear regression test (SPSS V23) reveals a significant statistical dependence of the retention of acetic acid (green marker) on pH, with p = 0.047. Acetic acid was not completely retained at pH 4.2 (R = 0.88), and an increase in retention was seen at higher pH. With increasing pH, the $H*$ also increases for acetic acid, (see Fig. S1). The retention coefficients for acetic acid were 0.81, 0.88, and 1.05 for pH values of 3.1, 4.2, and 7.0, respectively, while their corresponding standard deviations were 0.18, 0.12, and 0.2."*

Formic acid:

Linear regression test for formic acid gave a p-value of 0.182, indicating no significant dependence for pH, also seen in Fig 2 in the manuscript.  Formic acid is already completely retained (R =1) at pH 4. As such, any increase in pH would not lead to an enhancement of the retention, even though $KH*$ for formic acid varies in a similar fashion to acetic acid (Fig A).

**Text has been added/changed in L211 as:**

*"Formic acid (blue marker) did not show any dependency on pH (p = 0.182). Formic acid is already completely retained at pH 4.1 (R= 1.01), and as such, any increase in pH would not lead to an enhancement of the retention, even though $H*$ for formic acid varies in a similar fashion to acetic acid (Fig. S1)"*

2-nitrophenol:

Interestingly, 2-nitrophenol (with p = 0.005) also showed statistically significant dependence of retention on pH, for our measured pH range from 3 to 6. This result for 2-nitrophenol is contradictory to the expected form of dependence of KH* on pH seen in Fig. A.  Here, a probable explanation is that the 2-nitrophenol is more dissociated at pH 6 than at pH 3 and 4. We calculated the fraction of deprotonated to protonated ions at pH 3, 4 and 6 for 2-nitrophenol. This ratio was found to be $7 \times 10^{-5}$, $7 \times 10^{-4}$, $7 \times 10^{-2}$, at pH 3, 4 and 6 respectively. This means that at pH 6, about 7% of 2-nitrophenol is present in deprotonated form. During the freezing process, deprotonated molecules must undergo protonation to achieve neutrality before they can be expelled from the drop. At pH 6, a higher proportion of molecules remain confined within the drop due to the requirement for proton recombination prior to volatilization and subsequent expulsion. This pH dependence for 2-nitrophenol is also in agreement with Borchers et al. (2024), where they measured retention coefficients of α-pinene oxidation products and nitro-aromatic compounds during riming for cloud droplets.

**Text has been added/changed in L214 as:**

*"2-nitrophenol (red marker) showed statistically significant dependence of retention on pH (p = 0.005), for our measured pH range. The retention coefficients of 2-nitrophenol at pHs of 3.2 and 4.4 and 6 were 0.90, 0.90 and 1.05, respectively, and their corresponding standard deviations were 0.08, 0.05 and 0.11. This result for 2-nitrophenol is contradictory to the expected form of dependence of H∗ on pH, as in Fig. S1. 2-nitrophenol is more dissociated at pH 6 than at pH 3.2 and 4.4. The fraction of deprotonated to protonated ions at pH 3.2, 4.4 and 6 for 2-nitrophenol was found to be $7 \times 10^{-5}$, $7 \times 10^{-4}$ and $7 \times 10^{-2}$, respectively. This means that at pH 6, about 7% of 2-nitrophenol is present in deprotonated form. During the freezing process, deprotonated molecules must undergo protonation to achieve neutrality before they can be expelled from the drop. At pH 6, a higher proportion of molecules remain confined within the drop due to the requirement for proton recombination prior to volatilization and their subsequent expulsion. This pH dependence for 2-nitrophenol is also in agreement with Borchers et al. (2024), for riming retention of cloud droplets"*

Summary:

Given the overall high retention for formic, acetic acid and 2-nitrophenol, the dependency on pH for raindrops might not be critical factor, as compared to cloud droplets. The ice shell formation remains the major contributing factor for high retention in raindrops.

**2nd part:**

We did not derive the parameters a and b mentioned in equation 4. Rather, the values are taken from Borchers et al. (2024). They used the retention values from their study, along with previously measured retention coefficients from von Blohn et al. (2011,2013) and Jost et al. (2017) also involving cloud droplets – and updated the parameters a and b in light of their findings. We plotted this updated fit alongside our present data for retention of raindrops for comparison purposes – and to show that the dependency of retention on solubility and dissociation (i.e. on H*) do not entirely hold true for mm sized drops.

Text has been changed in L248 to better clarification as:

*"The relation between effective Henry's law coefficient and retention coefficient for cloud droplets i.e., retention-riming, was modeled by the following equation:"*

In L251 as:

*"Values a and b were taken from Borchers et al. (2024)."*

And in L255 as:

*"Equation 4 was plotted in Fig. 4 against our current data for comparing the dependency of R on H∗, for µm sized droplets and mm sized drops."*

**Editor: 3)** There are several studies that revealed that the gas-aqueous partitioning of nitrophenols in cloud droplets may not adhere to their Henry's law constants, e.g.

Lüttke et al. Phenols and Nitrated Phenols in Clouds at Mount Brocken, Intern. J. Environ. Anal. Chem..Vol. 74(1-4). pp. 69-89 Lüttke et al. Phase partitioning of phenol and nitrophenols in clouds, Atmos. Environm., 1997, 2649-55.

I wonder if the behavior of nitrophenol in your study could be partially explained by this.

**Response:**

Thank you for the comment and the references.

We did go through the suggested studies - Lüttke et al.(1997 and 1999).

Lüttke et al. (1997) showed that liquid-gas partitioning coefficient for 2-nitrophenol is about 6 times higher for their observations. A possible explanation was due to the adsorption of 2-nitrophenol on the surface of the droplets. They could not give satisfactory explanation for this increase. In Lüttke et al. (1999), they found that 2-nitrophenol in liquid phase can be approximately described by H*. In both these studies,

they refer to measured H* values for 2-nitrophenol from Tremp et al. (1993) and Schwarzenbach et al. (1988). More recent measurements (eg. Guo and Brimblecombe, (2007) in Sanders (2023)) for 2-nitrophenol show 2-fold higher H* compared to the above mentioned earlier measurements cited in Lüttke et al. (1997 and 1999) studies. Also, Lüttke et al. (1997 and 1999) measured droplets transported over time at mountain ranges. These carried over droplets could have other dissolved substances as well, which also might have had an influence on their results.

Retention of 2-nitrophenol has also been studied for riming retention, which follows the sigmoidal dependence on H* in Borchers et. al. (2024). They reported a low retention coefficient of 0.12 for 2-nitrophenol at pH 4. However, in our case, we found the longer solute expulsion timescale compared to the ice shell formation time leads to the high retention values observed for 2-nitrophenol, and perhaps not the influence of H*.

**Minor/Technical comments**

Please carefully proofread the paper. In particular pay attention to the correct use of articles.

**Editor: a)** I list a few places below where 'the' or 'a' is missing, e.g.

l. 20: from the boundary layer

l. 36: in the context

l. 331: could have the potential

l. 333: cracking of the ice shell

l. 337: where the fraction of liquid freezes and the majority …

**Editor: b)** Also please pay attention to the consistency of singular/plural forms of subject and verb, e.g.

**Editor: l. 61:** Freezing of raindrops is…

**Editor: Table 2:** …temperature was

**Editor: l. 323:** …rates … imply

**Response:**

Thank you for pointing out these small yet very important grammatical errors. Changes have been made accordingly in the above mentioned points.

**Editor: Abstract:l. 1/ 2:** You may want to consider improving the first sentence (in particular since it is the first sentence), *The interaction with freezing processes and*

*vertical transport of trace gases into the upper atmosphere during deep convection is critical to understanding the distribution of aerosol precursors and their climate effects.*

1) 'Interactions with..' does not seem right here

2) Processes do not really 'interact' – they may be coupled or influence/affect each other.

If I understand correctly, you want to say "Freezing processes affect the vertical transport of trace gases into the upper troposphere...

**Response:**

Thank you for pointing it out. Text has been changed in L1 as:

*"The influence of freezing processes and vertical transport of trace gases..."*

**Editor: l. 10:** *"Thus, for rain sized drops almost everything is fully retained during the freezing process, even for species with low effective Henry's law constants."*

This sentence sounds quite colloquial. Given that you define 'retention coefficient', it may be clearer or more precise to say that the retention coefficients for all single compounds and mixtures were near 1 (or give a range)

Can you specify 'low effective Henry's law constant'?

**Response:**

Text has been changed in L9 as:

*"Thus, for rain sized drops almost everything is fully retained during the freezing process i.e., retention coefficients close to 1, even for species with low effective Henry's law constants, $H^* < 10^{-4}$."*

**Editor:l. 27:** 'evident' seems redundant here

**Response:** The word 'evident' has been removed in L27

**Editor:l. 30:** Here you use 'drop' in the context of clouds – given the referee comment and your response, shouldn't it be 'droplet'?

**Response:**

Here we introduced the definition of retention coefficient in a general sense. We do see the fact that it might lead to confusion. Text has been changed in L28-31 as:

*"Trace gases dissolved in these droplets could be either retained, revolatized, or scavenged during the freezing process (Pruppacher and Klett, 2010). The fraction of chemical species remaining inside the frozen hydrometeor, compared to their initial concentration in liquid phase before freezing, results in the so-called retention coefficient."*

**Editor:l. 32, 37** (and maybe other places in the manuscript): For better readability, please move the references to the end of the sentence.

**Response:**

Thank you for pointing it out. References have been moved to the end of the sentences in L32, 37 and other places as well.

**Editor:l. 43:** 'Additionally' implies that H* is neither a chemical nor physical property as they were already mentioned in the previous sentence. Thus, 'additionally' seems redundant here.

**Response:** The word 'additionally' has been removed in L43.

**Editor:l. 55:** 'A significant difference from a physical perspective in terms of retention of trace gases for cloud droplets and rain drops would be the initiation and pathway of freezing'

- Why do you use subjunctive ('would')? If it is a well-known fact, 'is' is appropriate. (Please check the full manuscript for use of 'would' and decide whether the use of indicative ('is') is appropriate. )

- 'initiation and pathway of freezing' - is usually referred to as 'freezing mechanism'.

**Response:**

Thank you for pointing it out. Text has been changed in L54 as:

*"A significant difference from a physical perspective in terms of retention of trace gases for cloud droplets and rain drops is the freezing mechanism."*

As per suggestion, text has been changed/modified for instances of 'would', changes can be seen in the track-changes document.

**Editor:l. 57:** 'was implemented' sounds odd. You may implement something in a model but this is certainly not meant here... isn't something like 'was the main mechanism' or 'took place' more appropriate?

**Response:**

Text has been changed in L56 as:

*"For riming experiments involving cloud droplets freezing is initiated upon contact with a frozen substrate, whereas, for rain drops investigated in this present study, immersion freezing was the main mechanism."*

**Editor:l. 67:** It may be useful to add the H* values here already, together with the pH value.

**Response:**

Text has been added in L65 as:

*"To visualize our experimental outlook, we selected four chemical substances namely: 2-nitrophenol, acetic acid, formic acid, and nitric acid, with increasing H\*values of $3.50 \times 10^3$, $1.28 \times 10^5$, $8.31 \times 10^5$ and $7.56 \times 10^{11}$, respectively, at 0 ◦C and pH about 4, for all."*

**Editor:l. 69:** what do you mean by '…values for riming with cloud droplet sizes' – is it simply 'in riming cloud droplets'?y

**Response:**

Text has been changed in L68 as:

*"These substances are commonly found in the atmosphere and their previously measured retention coefficient values in riming cloud droplets lie between 0 to 1 and scale with H\*."*

**Editor:l. 104:** - ppm and ppb are mixing rations, not concentrations.

- Please specify that you mean 'ppb to ppm (on mass basis)' to make clear that you mean 1 g in 10^9 g or 10^6 g, i.e. 1 ug/L or 1 mg /L (assuming that water density = 1 g/cm3)

This avoids confusion since gas phase mixing ratios of trace gases are commonly given in ppb whereas e.g 50 ppb ozone means '50 molecules out of 10^9 molecules'

**Response:**

Thank you for pointing it out. Text has been changed in L102 as:

*"Typical mixing ratio of dissolved gases in the atmosphere lies in the range of ppb to tens of ppm (on mass basis)"*

**Editor:l. 111:** 'least' should be 'lowest'

**Response:**

Text has been changed in L110 as:

*"- which has the lowest molar mass among the investigated species – "*

**Editor:l. 120:** 'benzoic' misspelled

**Response:** Thank you for pointing it out. Spelling correction has been made for *"2-nitrobenzoic acid"* in L119.

**Editor:l. 141:** You used D already for drop diameter (abstract). I suggest changing it there and simply spell out 'drop diameter: 2 mm).'

**Response:** Thank you for your suggestion. Changes have been made in L3 as:

*"...for freely levitating rain drops (drop diameter: 2 mm) using an..."*

**Editor:l. 164/166:** Even though Referee #1 did not specifically comment on the text in these lines, 'average freezing temperature' should be also replaced here by 'median freezing temperature'.

**Response:** Thank you for pointing it out. Changes have been made in L162 and L164.

average freezing temperature replaced with *"median freezing temperature."*

**Editor:l. 169:** Please clarify this sentence: " The 50% frozen fraction at-23∘C was found to be −6.9±1.1∘C."

**Response:**

"50% frozen fraction" has been used to synonymously refer to the median freezing temperature, as stated in L161. However, to avoid any further confusion the term "50% frozen fraction" has been replaced with median freezing temperature.

Changes have been made in L161 as:

*"From the temperature profile obtained for experiments conducted at -15 $^0$C cold room temperature and 0.2 g/L AgI, the median drop freezing temperature was found to be −3.9 ± 0.3 $^0$C, under these experimental conditions (Fig. A2)."*

and in L166 as:

*"The median drop freezing temperature for -23 $^0C$ cold room temperature was found to be −6.9 ± 1.1 $^0C$."*

**Editor:l. 171:** 'sized' can be omitted here and also in the remainder of the manuscript for similar instances.

**Response:** Thank you for your suggestion. The word 'sized' has been removed in L168 and 3 other instances with similar context.

**Editor:Table 2:** Clarify in the caption whether the R values are averaged over all pH values or only apply to a specific pH.

**Response:**

Table 2 caption has been changed as:

*"Retention coefficients at drop freezing temperature of −3.9 ± 0.3 $^0C$ and pH values about 4 for all the investigated substances. The corresponding walk-in cold room temperature (ambient temperature) was −15 ± 1 $^0C$."*

**Editor:l. 182:** "Brand (2014) studied the retention of large drops (2.67 mm and 7.25 mm spherical equivalent diameter)" – please clarify what Brand investigated. It should be the retention of gases (organic acids? All the same compounds as you used in the present study?) in large drops.

**Response:**

Brand studied the retention of formic, acetic, oxalic and malonic acids. Text has been added in L179 as:

*"Brand (2014) studied the retention of formic, acetic, oxalic and malonic acids – for large drops (2.67 mm and 7.25 mm spherical equivalent diameter) by freezing them on a Teflon coated pallet – also reported high retention coefficients (close to 1)"*

**Editor:l. 185:** 'with which ...was realized' can replaced 'representing'

**Response:**

Text has been changed in L182 as:

*"However, in our study contact-free immersion freezing was employed, representing a more realistic scenario to initiate freezing as compared to Brand (2014)."*

**Editor:l. 193"** replace 'least' by 'lowest'

Text has been changed in L190 as:

*"...having the lowest H* among the investigated substance..."*

**Editor:l. 209-215:** What is the main message here?

First you say that *"Acetic acid (green marker) and formic acid (blue marker) did not show any apparent dependency on pH"*

Then you say that "*The retention coefficients for acetic acid were 0.81, 0.88, and 1.05 for pH values of 3.1, 4.2, and 7.0, respectively, while their corresponding standard deviations were 0.18, 0.12, and 0.2"* – doesn't this trend show a dependence? I understand your argument that the standard deviations are larger than the differences between the mean values – however, yet, the figure shows a clear trend and an average R at pH = 7 that is about 25% higher than that at pH = 3.1.

*"From Fig. 2a, one can infer a slight dependency on pH for 2-nitrophenol, and almost none for acetic acid and formic acid."*

I see it the opposite way based on the figure, i.e. that there is barely any dependence of R on pH for nitrophenol (at least at pH < 5); however, there is a steady increase of R with pH for acetic acid.

I understand your argument that your conclusions are based on the results of 11 experiments. Why don't you show these values rather than just the averages ± standard deviation that (falsely?) imply a trend and therefore contradict your text?

**Response:** We are thankful for your insightful comment.

In our response to "Main editor comments: **no.2, 1st part**, we have addressed this concern extensively and added new text to the manuscript accordingly for better clarity. Kindly refer to our response for the highlighted section.

Below we provide the plot showing our measured retention coefficients at different pH values for the single components.

[Figure]

The overlap of the data points doesn't seem very neat and informative in the new plot. As such, we would prefer to include the original plot for Fig 2 in the manuscript, with the averages ± standard deviation, provided there aren't any further objections. We have provided this figure in the supplement as Figure S2, for reference.

**Editor:l. 219 - 224:** It is difficult to understand what you are saying here.

*"pH of the solutions were altered by adding HCl and NaOH, which could also interact with the investigated substances and dissociate them into their ionic form"*

Isn't this idea of pH adjustment that you change the proportions of dissociated vs undissociated forms? I suggest omitting this sentence as the second part is confusing (if not even wrong as adding NaOH does not lead to dissociation but association of H+ and carboxylates), and the first part was already mentioned in Section 2.

*" In this case the overall concentration of the investigated substances could be lowered."* Which concentration is lowered under what conditions? When acids dissociate (i.e. at enhanced pH) the total aqueous phase concentration (acid + anion) actually increases.

Or are you saying that the solubility of the solutes is expected to decrease in the presence of additional solutes such as HCl and NaOH due to salting-out effects? Are there any references for this? In such a case, the Henry's law constants for pure water may not be applicable.

*"After addition, the lowest measured initial liquid phase concentration was 17.8 mg/L (11% decrease)."* Is this an expected trend or is this random variation due to evaporation of acids?

**Response:**

Thank you for pointing it out. Perhaps these sentences might create more confusion than clarity to the readers. We also did not refer to the salting effects either. We meant to say that the addition of HCl and NaOH didn't create a bias in terms of concentration in our measurements. We measured the solute concentration before freezing, after freezing and desorption, for calculating retention coefficients. So, the changes in mass concentrations are accounted for already. As such, the referred texts in L219-224 have been removed altogether to avoid any further confusion. The changes can be seen in the track-changes document.

**Editor:Table 3:** Please indicate that you use dimensionless Henry's law constants.

**Response:**

Text has been changed in Table 3 as:

*"Dimensionless Henry's law constant" in place of previously* written "Effective Henry's law constant"

**Editor:l. 325/6:** '*from their numerical simulations'* seems at a wrong place in the sentence. Please clarify.

**Response:**

Text has been changed for better clarity in L326 as:

*"Stuart and Jacobson (2006) reported the formation of liquid pockets that can trap solutes during freezing, informed from previous studies of dendritic crystal growth in solutions."*

**Editor:l. 343:** "*Our results show higher retention coefficients close to 1 for mm sized raindrops for similar substances from previously studied retention coefficients"*

This sentence should be restructured for clarity, e.g. Our results show higher retention coefficients (close to 1) for similar substances in mm sized raindrops as compared to previously determined retention coefficients in um sized cloud droplets.

**Response:**

Thank you for your suggestion. Text has been changed in L365 as:

*"Our results show higher retention coefficients (close to 1) for similar substances in mm sized raindrops as compared to previously determined retention coefficients in μm sized cloud droplets (von Blohn et al. 2011; Jost et al. 2017; Borchers et al. 2024)"*

**Editor:Section 4:** Please make sure that the conclusion section adheres to the author guidelines at https://www.atmospheric-chemistry-and-physics.net/policies/guidelines_for_authors.html

**Response:**

Thank you for pointing it out. We have added a summary section, which was missing in our conclusions. And rearranged the conclusions as per the guidelines provided.

Text has been added in L350 as:

*"At the onset, we successfully characterized the freezing of levitated rain drops (2.0 ± 0.1mm) at three different concentrations and temperatures using the acoustic levitator setup. We measured the retention coefficients of nitric acid, formic acid, acetic acid and 2-nitrophenol as single components and their combinations as binary mixtures, during the freezing of rain drops. In addition to these measurements, we also checked the sensitivity at three different pH levels (pH 3, 4 and 6/7) and at two different temperatures (−3.9 ± 0.3∘C and −6.9 ± 1.1∘C)."*

**References:**

Borchers, C., Seymore, J., Gautam, M., Dörholt, K., Müller, Y., Arndt, A., Gömmer, L., Ungeheuer, F., Szakáll, M., Borrmann, S., et al.:Retention of α-pinene oxidation products and nitro-aromatic compounds during riming, Atmospheric Chemistry and Physics, 24, 13 961–13 974, 2024.

Guo XX, Brimblecombe P. Henry's law constants of phenol and mononitrophenols in water and aqueous sulfuric acid. Chemosphere. 2007 Jun;68(3):436-44. doi: 10.1016/j.chemosphere.2007.01.011. Epub 2007 Mar 6. PMID: 17343895.

Jost, A., Szakáll, M., Diehl, K., Mitra, S. K., and Borrmann, S.: Chemistry of riming: the retention of organic and inorganic atmospheric trace constituents, Atmospheric Chemistry and Physics, 17, 9717–9732, 2017.

R. Sander: Compilation of Henry's law constants (version 5.0.0) for water as solvent, Atmos. Chem. Phys., 23, 10901-12440 (2023), doi:10.5194/acp-23-10901-2023

Schwarzenbach, R. P., Stierli, R., Folsom, B. R., & Zeyer, J.: Compound properties relevant for assessing the environmental partitioning of nitrophenols, Environ. Sci. Technol., 22, 83–92, doi:10.1021/ES00166A009 (1988).

Tremp, J., Mattrel, P., Fingler, S., & Giger, W.: Phenols and nitrophenols as tropospheric pollutants: Emissions from automobile exhausts and phase transfer in the atmosphere, Water Air Soil Pollut., 68, 113–123, doi:10.1007/BF00479396 (1993)

von Blohn, N., Diehl, K., Mitra, S., and Borrmann, S.: Wind tunnel experiments on the retention of trace gases during riming: nitric acid, hydrochloric acid, and hydrogen peroxide, Atmospheric Chemistry and Physics, 11, 11 569–11 579, 2011.

von Blohn, N., Diehl, K., Nölscher, A., Jost, A., Mitra, S. K., and Borrmann, S.: The retention of ammonia and sulfur dioxide during riming of ice particles and dendritic snow flakes: laboratory experiments in the Mainz vertical wind tunnel, Journal of Atmospheric Chemistry, 70, 131–150, 2013.